# A mathematical model of potassium homeostasis: Effect of feedforward and feedback controls

**Melissa M. Stadt**[1]*, **Jessica Leete**[2], **Sophia Devinyak**[3], **Anita T. Layton**[1,4,5,6]

**1** Department of Applied Mathematics, University of Waterloo, Waterloo, Ontario, Canada, **2** Computational Biology and Bioinformatics Program, Duke University, Durham, North Carolina, United States of America, **3** Department of Physics and Astronomy, University of Waterloo, Waterloo, Ontario, Canada, **4** Cheriton School of Computer Science, University of Waterloo, Waterloo, Ontario, Canada, **5** Department of Biology, University of Waterloo, Waterloo, Ontario, Canada, **6** Department of Pharmacy, University of Waterloo, Waterloo, Ontario, Canada

* mstadt@uwaterloo.ca

**Data Availability Statement:** The source code and data used to produce the results and analyses presented in this manuscript are available on a GitHub repository at https://github.com/Layton-Lab/Kregulation. We have also used Zenodo to

## Abstract

Maintaining normal potassium ($K^+$) concentrations in the extra- and intracellular fluid is critical for cell function. $K^+$ homeostasis is achieved by ensuring proper distribution between extra- and intracellular fluid compartments and by matching $K^+$ excretion with intake. The $Na^+$-$K^+$-ATPase pump facilitates $K^+$ uptake into the skeletal muscle, where most $K^+$ is stored. $Na^+$-$K^+$-ATPase activity is stimulated by insulin and aldosterone. The kidneys regulate long term $K^+$ homeostasis by controlling the amount of $K^+$ excreted through urine. Renal handling of $K^+$ is mediated by a number of regulatory mechanisms, including an aldosterone-mediated feedback control, in which high extracellular $K^+$ concentration stimulates aldosterone secretion, which enhances urine $K^+$ excretion, and a gastrointestinal feedforward control mechanism, in which dietary $K^+$ intake increases $K^+$ excretion. Recently, a muscle-kidney cross talk signal has been hypothesized, where the $K^+$ concentration in skeletal muscle cells directly affects urine $K^+$ excretion without changes in extracellular $K^+$ concentration. To understand how these mechanisms coordinate under different $K^+$ challenges, we have developed a compartmental model of whole-body $K^+$ regulation. The model represents the intra- and extracellular fluid compartments in a human (male) as well as a detailed kidney compartment. We included (i) the gastrointestinal feedforward control mechanism, (ii) the effect of insulin and (iii) aldosterone on $Na^+$-$K^+$-ATPase $K^+$ uptake, and (iv) aldosterone stimulation of renal $K^+$ secretion. We used this model to investigate the impact of regulatory mechanisms on $K^+$ homeostasis. Model predictions showed how the regulatory mechanisms synthesize to ensure that the extra- and intracelluller fluid $K^+$ concentrations remain in normal range in times of $K^+$ loading and fasting. Additionally, we predict that without the hypothesized muscle-kidney cross talk signal, the model was unable to predict a return to normal extracellular $K^+$ concentration after a period of high $K^+$ loading or depletion.

assign a DOI to the repository: 10.5281/zenodo.
7308265.

**Funding:** This work is supported by the Canada
150 Research Chair program and by the National
Science and Engineering Research Council of
Canada via a Discovery award (RGPIN-2019-
03916) to A.T. L and a Canada Graduate
Scholarship (CGS-D) to M.M.S. The funders had
no role in study design, data collection and
analysis, decision to publish, or preparation of the
manuscript.

**Competing interests:** The authors have declared
that no competing interests exist.

## Author summary

Potassium ($K^+$) homeostasis is crucial for normal cell function. Dysregulation of $K^+$ can
have dangerous consequences and is a common side effect of pathologies, medications, or
changes in hormone levels. Due to its complexities, how the body maintains extra- and
intracellular $K^+$ concentrations each day is not fully understood. Of particular interest is
capturing how regulatory mechanisms synthesize to be able to keep extracellullar $K^+$ con-
centration within a tight range of 3.5–5.0 mEq/L. There are a multitude of physiological
processes involved in $K^+$ balance, making its study well suited for investigation using
mathematical modeling. In this study, we developed a compartment model of extra- and
intracellular $K^+$ regulation including the various regulatory mechanisms and a detailed
kidney model. The significance of our research is to quantify the effect of individual regu-
latory mechanisms on $K^+$ homeostasis as well as predict the potential impact of a hypothe-
sized signal: muscle-kidney cross talk.

## Introduction

Potassium ($K^+$) plays an essential role in maintaining normal cellular function. In particular,
the transmembrane $K^+$ gradient maintains the transmembrane potential gradient, which is
critical for cellular function [1]. In electrically excitable cells, such as neurons and muscle cells,
the transmembrane potential gradient is used for transmitting signals between different parts
of a cell [2]. Muscle cells, for instance, rely on the gradient to drive action potentials to facilitate
muscle contractility [2]. The transmembrane potential gradient also impacts transporter and
ion channel activity [1, 2]. In humans, normal extracellullar $K^+$ concentration is about 3.5–5.0
mEq/L; intracellular $K^+$ concentration is about 120–140 mEq/L [1, 3]. To achieve $K^+$ homeo-
stasis, $K^+$ excretion must first match intake in the long run to avoid net accumulation or loss;
then the body's $K^+$ must be properly distributed between extra- and intracellular fluids [4].

When extracellular $K^+$ concentrations are outside the tight range of 3.5–5.0 mEq/L, cellular
balances of $Na^+$, $K^+$, and $Ca^{2+}$ are altered, which can have dangerous consequences [1, 2]. The
distribution of $K^+$ between the extra- and intracellular fluid, as well as renal $K^+$ handling
(which impacts total body $K^+$) can be altered by hormones, drugs, various pathological states,
or even just a simple meal. Uncompensated $K^+$ imbalance or dyskalemia (either hyper- or
hypokalemia) can lead to constipation, muscle weakness, fatigue, and cardiac arrhythmias,
and in severe cases could be fatal. Hypokalemia can be a result of insufficient dietary $K^+$ or
excessive $K^+$ excretion [4, 5]. The major causes of hyperkalemia include chronic kidney dis-
ease, uncontrolled diabetes, adrenal disease, consuming excessive dietary $K^+$, and some medi-
cations [2, 4].

$K^+$ homeostasis has received much attention, from the research community as well as the
general public, because not only can large fluctuations in the extracellular $K^+$ concentrations
cause potentially life-threatening consequences, maintaining the extracellular $K^+$ concentra-
tion within a small range is also a daily homeostatic challenge. First, unless an opposing trans-
membrane potential is sustained, the large transmembrane $K^+$ concentration gradient (noted
above) would drive a massive $K^+$ flux out of the cells. That could produce a substantial pertur-
bation in extracellular $K^+$ concentration, given that the extracellular pool is much smaller than
the intracellular pool (70 compared to 3,500 mEq [1, 3]). A second challenge is that $K^+$ has
the highest ratio of daily intake to extracellular pool size of all the major electrolytes: recom-
mended daily intake of $K^+$ is 120 mEq [1], about 70% more than the extracellular $K^+$ pool. (In
comparison, the recommended daily intake of $Na^+$ is 2,300 mEq, which is only slightly higher

than its extracellular pool of about 2,100 mEq [6].) As one can imagine, a simple meal could have nearly as much $K^+$ as in the extracellular fluid! As such, without rapid cellular $K^+$ uptake, a meal could cause drastic changes in extracellular fluid $K^+$ concentration. Finally, since about 90% of the $K^+$ excretion occurs via urine (remaining via feces), urinary excretion of $K^+$ must be nearly equal to $K^+$ intake. Because $K^+$ intake varies widely each day, the kidneys must be capable of varying urinary $K^+$ excretion to match varied $K^+$ intake to avoid excessive $K^+$ retention or loss.

Mammals have evolved to develop a number of regulatory mechanisms that target renal function and cellular $K^+$ uptake. *How do these mechanisms interact? What role does each of these mechanisms play under different types of $K^+$ perturbations (e.g., a $K^+$-rich meal, repeated $K^+$ loading, or $K^+$ depletion)? Given the potentially and immediate dire consequences of hypo- and hyperkalemia, how can the kidneys sense that extracellular $K^+$ may be in danger of going out of range before it happens, and promptly adjust $K^+$ excretion appropriately?* Indeed, understanding the big picture of $K^+$ homeostasis is an open area of study with significant pieces of the puzzle that have yet to be put together. Knowledge gaps primarily lie in understanding how signals and sensors on both cellular uptake and renal $K^+$ handling synthesize to maintain an extracellular fluid $K^+$ concentration within tight range [1]. Below we give a brief overview of the individual mechanisms involved in $K^+$ regulation. For more details we refer the readers to Refs. [1, 3, 4, 7].

## Kidney function

In the long term, the kidney is the principal defense against chronic $K^+$ imbalance. Potassium is freely filtered through the glomerulus and into the nephrons. As the luminal fluid passes through the nephrons, various regulatory processes control how much $K^+$ is secreted or reabsorbed [7–9]. A schematic of renal $K^+$ handling is shown in Fig 1. In a male rat, the proximal tubules consistently reabsorb about two-thirds of filtered $K^+$ and then the loops of Henle reabsorb 20–25% more so that together these segments reabsorb around 90% of the filtered $K^+$ [7]. This leaves the fine tuning to the downstream segments, namely distal tubules and collecting ducts.

The distal tubular segments have the ability to secrete or reabsorb $K^+$, depending on the state of the body. During normal or high $K^+$ intake, the distal tubular segments secrete 10% to 50% of filtered $K^+$ [7]. Secretion rate is largely controlled by aldosterone (ALD), which increases the luminal permeability of $K^+$ along the distal tubular segments. However, the distal tubule's capacity to reabsorb $K^+$ is limited (up to 3% of filtered $K^+$ in male rats), meaning that in times of drastically low $K^+$ intake, it is up to the downstream collecting duct to conserve $K^+$ via reabsorption [7, 8].

The cortical collecting duct and outer medullary collecting duct consist of the principal cells and intercalated cells, which are responsible for $K^+$ secretion and reabsorption, respectively. Principal cells comprise approximately 70–75% of collecting duct cells in rats [10]. These cells use the electrochemical gradient established by $Na^+$ entry into the cell to drive $K^+$ secretion. The rate of $K^+$ secretion is regulated by dietary $K^+$ intake, intracellular $K^+$ concentration, $Na^+$ uptake into the cells, urine flow rate, and hormones such as ALD and angiotensin II [7, 8]. Through upregulation of $H^+$-$K^+$-ATPase in the intercalated cells, the collecting duct can reabsorb up to 10% of filtered $K^+$ [7, 9].

## Internal $K^+$ balance

While the kidney determines overall $K^+$ conservation and loss, it takes time to have an effect [1, 11]. After a $K^+$ load, in the first 3–6 hours, only about 50% of the $K^+$ load is excreted via

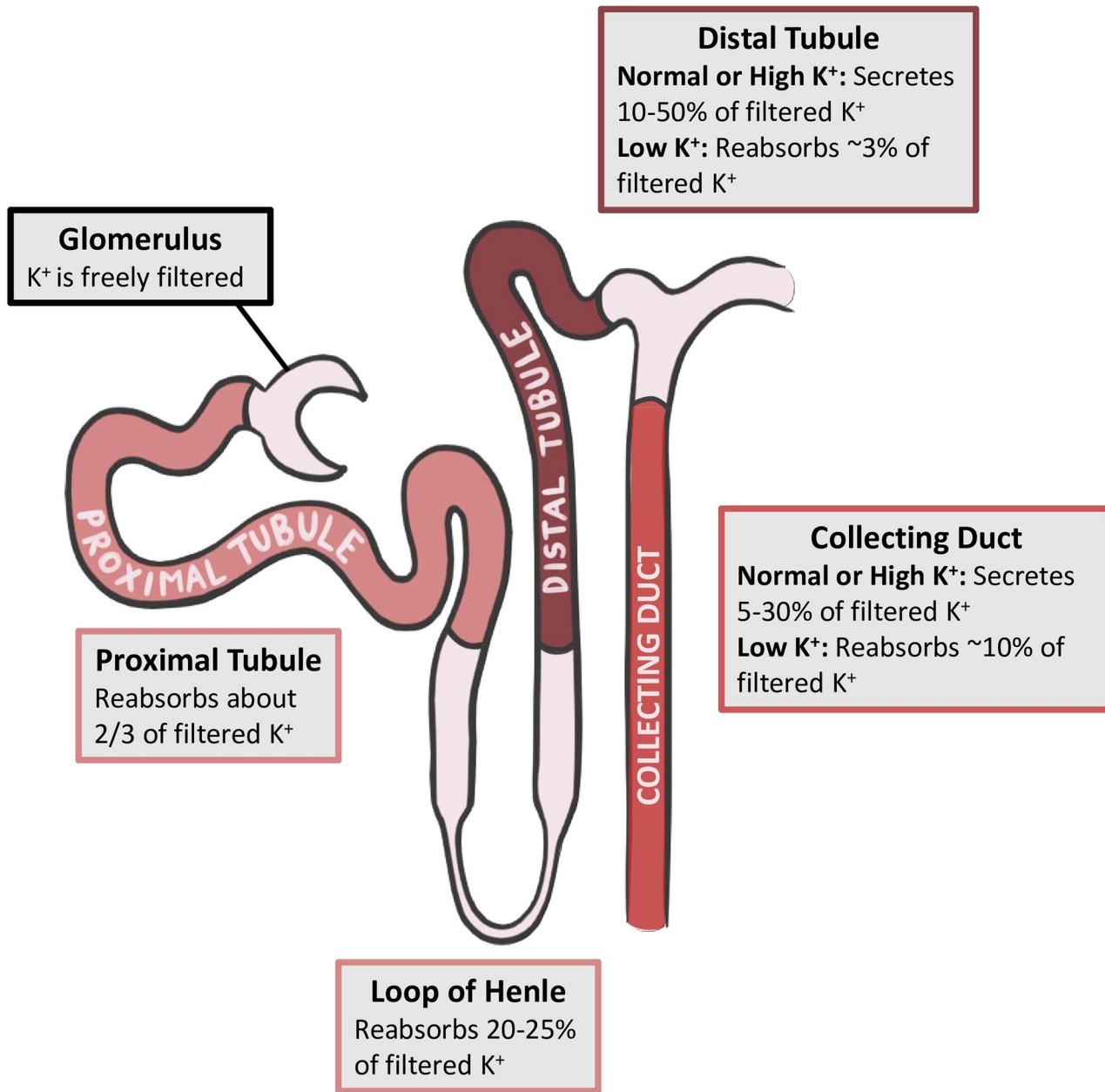

**Fig 1. Renal handling of K⁺.** Transport of K⁺ in key nephron segments during normal, low, and high K⁺ intake.

urine; the remaining K⁺ load is excreted over the next 10 hours [1, 12]. This time scale gives rise to the question: *Following a large K⁺ intake (e.g., a K⁺-rich meal), how can the low extracellular K⁺ concentration be maintained?* This is an important question because even transient rises in extracellular K⁺ concentration can be dangerous. If the remaining half of the K⁺ load (not excreted in urine) stayed in the extracellular fluid, plasma K⁺ concentration would rise significantly, potentially to dangerous levels. To avoid that scenario, the body rapidly increases uptake of K⁺ into the cells, primarily, the skeletal muscle cells. This quick storage of K⁺ allows the body to maintain an appropriately low extracellular K⁺ concentration, and to later slowly release K⁺ back into the bloodstream without large perturbations in concentration. Cellular

uptake of $K^+$ can be enhanced by the activation of $Na^+$-$K^+$-ATPase pumps by insulin and ALD [13, 14].

## Feedback and feedforward controls

A number of feedback and feedforward controls are involved in $K^+$ homeostasis by targeting either cellular $K^+$ uptake or renal function. Feedback controls respond to changes in extracellular $K^+$ concentration. High extracellular $K^+$ concentration increases ALD secretion. As mentioned previously, ALD impacts $K^+$ regulation by increasing $K^+$ secretion into the distal segments of the nephrons, as well as increasing cellular $K^+$ uptake via $Na^+$-$K^+$-ATPase pump stimulation [1, 13].

Feedforward control refers to a pathway that reacts to an environmental input in a predetermined way, without reference to the value being controlled. A typical meal consists of both glucose and $K^+$. Glucose intake stimulates insulin secretion, which then stimulates cellular uptake of $K^+$. $K^+$ intake has been shown to activate a feedforward control in which distal tubule $K^+$ secretion is stimulated before changes in plasma $K^+$ concentration or ALD concentration can be observed [1, 15–18]. This signal will be referred to as the *gastrointestinal feedforward control mechanism*.

Identifying all the $K^+$ regulatory mechanisms remains an open area of research [1]. For example, following a significant elevation in total body $K^+$, how does the body safely get rid of the excess $K^+$ without significantly, and potentially dangerously, raising extracellular $K^+$ levels? One hypothesized mechanism is *muscle-kidney cross talk*, whereby intracellular $K^+$ concentration of skeletal muscle cells can directly affect urine $K^+$ excretion without first causing changes in extracellular $K^+$ concentration [1, 14, 19].

With the multitude of physiological processes involved in $K^+$ balance, mathematical modeling is well suited for studying their interactions. Pettit and Vick [20] presented a two-compartment model to analyze data from dogs to understand the contribution of insulin to extrarenal $K^+$ homeostasis. Their model represents $K^+$ storage in the extra- and intracellular compartments, together with their $K^+$ exchange. A similar model was developed by Rabinowitz [21] to analyze data in sheep and understand the role of a gastrointestinal feedforward control mechanism. Youn et al. [22] used two-compartment and three-compartment models to analyze fluxes of $K^+$ between the extracellular and intracellular space as well as $K^+$ uptake by red blood cells. Maddah and Hallow [23] developed a quantitative systems pharmacology model that captures the effect of aldosterone on $K^+$ homeostasis to simulate the effects of spironolactone treatment in patients with hyperaldosteronism.

To unravel the roles of the multitudes of organ systems and regulatory mechanisms involved in maintaining $K^+$ balance, we built upon the above studies and developed a more comprehensive computational model of $K^+$ homeostasis. We apply the model to investigate how each of the feedback and feedforward controls contributes to the regulation of plasma $K^+$ concentration, to the rapid excretion of $K^+$ following a $K^+$-rich meal, and to the preservation of $K^+$ during periods of $K^+$ depletion. We also simulate muscle-kidney cross talk and predict how this hypothesized mechanism may impact recovery from repeated $K^+$ loading or $K^+$ depletion.

## Materials and methods

We developed a compartmental mathematical model of $K^+$ regulation in a man. (We refer to this as a model of a man, instead of more generally as a human model, due to the known sex differences in renal $K^+$ handling [24–26], ALD concentration [27], and plasma $K^+$ concentration [28], all of which affect $K^+$ homeostasis. Parameters for the present models were fitted for

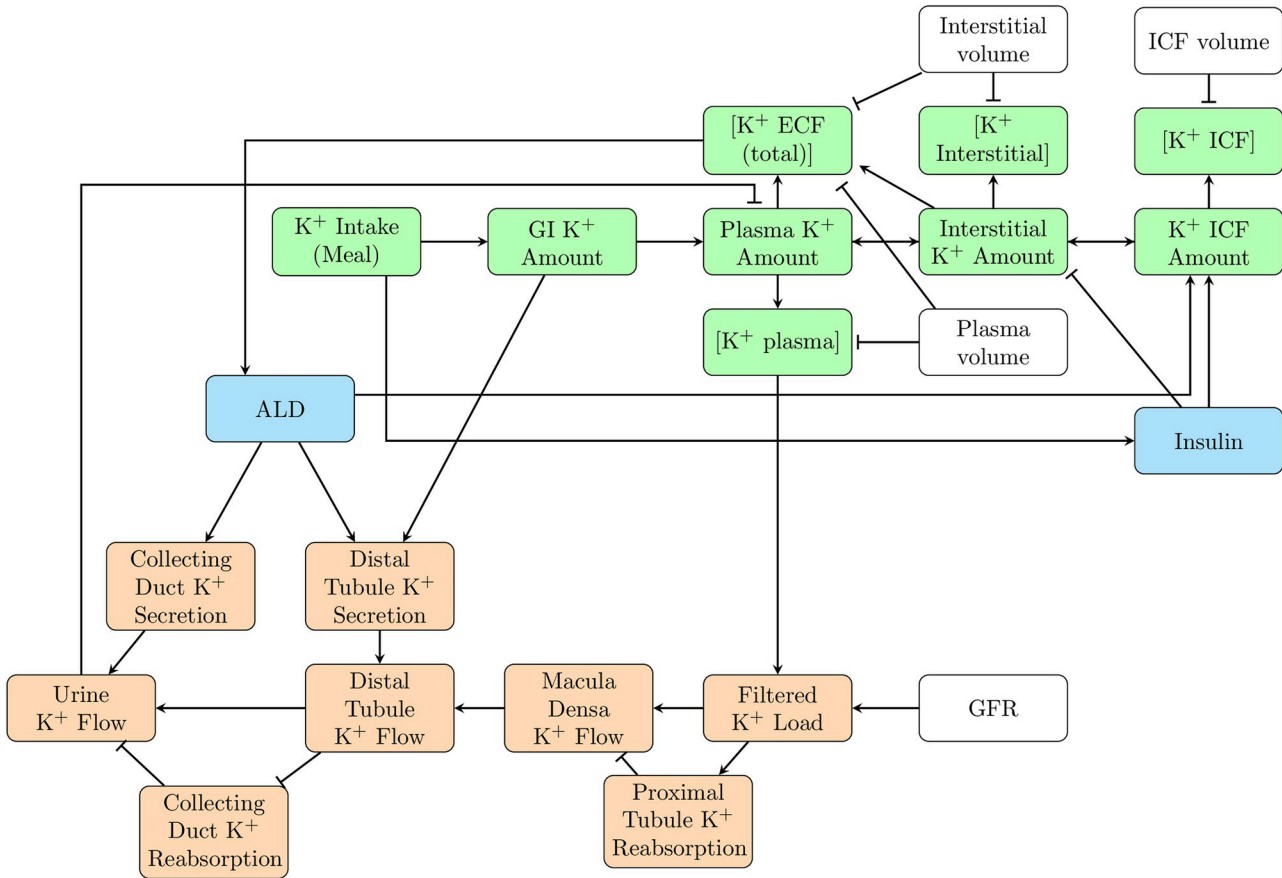

**Fig 2. Model schematic.** Flow chart depicting our mathematical model of K⁺ regulation. Green nodes represent internal K⁺ balance; orange, kidney function; blue, hormones (i.e., ALD and insulin); and white, constant model parameters. Pointed arrows indicate stimulation; blunted arrows indicate inhibition. ALD: aldosterone, GFR: glomerular filtration rate, GI: gastrointestinal, Interstitial: interstitial fluid, ICF: intracellular fluid, ECF: extracellular fluid

a man, not a woman.) The model represents the gut and hepatoportal circulation, intracellular fluid, extracellular fluid, and the kidney. The extracellular fluid compartment is divided into two sub-compartments: blood plasma and interstitial fluid. The physiological processes that are represented include dietary K⁺ intake, delivery of K⁺ through the hepatoprotal circulation to the blood plasma, intra- and extracellular K⁺ exchange, renal K⁺ transport, the gastrointestinal feedforward control mechanism, the effect of insulin on Na⁺-K⁺-ATPase K⁺ uptake, and feedback control of ALD on both Na⁺-K⁺-ATPase and distal segment K⁺ secretion. A flow chart depicting the system is shown in Fig 2. Parameter values are given in Table 1.

## Internal K⁺ balance

Oral K⁺ intake ($\Phi_{Kin}$) is treated as an independent variable. By varying $\Phi_{Kin}$, one can study the model's response to changes in K⁺ intake. For instance, if $\Phi_{Kin}$ is set to a constant ($\Phi_{Kin}^{ss}$), one may obtain a steady state solution to the model equations. Realistically, K⁺ intake varies throughout the day, with periods of high intake (meals) and periods of no intake (between meals). Thus, K⁺ intake is best approximated by a time-dependent function, as done in the simulations shown in Results.

**Potassium intake.** When $K^+$ is consumed orally (i.e., a meal), about 90% of the $K^+$ is absorbed by the intestine, then goes through the hepatoportal circulation into the blood plasma of the general circulation. The remaining 10% is excreted through feces. The amount of $K^+$ in the gut and hepatoportal circulation is denoted by $M_{Kgut}$ and is determined by

$$\frac{dM_{Kgut}}{dt} = 0.9 \times \Phi_{Kin} - k_{gut} \times M_{Kgut} \tag{1}$$

where $k_{gut}$ is a parameter that determines how fast the $K^+$ is delivered to the plasma.

**Extracellular fluid.** The extracellular fluid is made up of the blood plasma and the interstitial fluid. As noted above, after $K^+$ is ingested and processed, it is absorbed into the blood. The amount of $K^+$ in the blood plasma ($M_{Kplasma}$) is given by

$$\frac{dM_{Kplasma}}{dt} = k_{gut} \times M_{Kgut} - \Phi_{ECF} - \Phi_{uK} \tag{2}$$

where $\Phi_{uK}$ denotes urine $K^+$ excretion; $\Phi_{ECF}$ denotes the diffusion of $K^+$ from the blood plasma to the interstitial space and is given by

$$\Phi_{ECF} = P_{ECF} \times (K_{plasma} - K_{interstitial}) \tag{3}$$

where $P_{ECF}$ is a permeability parameter (see Table 1), $K_{plasma}$ denotes the plasma $K^+$ concentration, and $K_{interstitial}$ denotes $K^+$ concentration in the interstitial fluid.

**Table 1. Model parameter values used in the baseline $K^+$ homeostasis mathematical models.**

| Parameter | Description | Value | Unit | Ref |
|---|---|---|---|---|
| $k_{gut}$ | $K^+$ delivery rate to plasma | 0.01 | $min^{-1}$ | |
| $P_{ECF}$ | effective $K^+$ permeability in extracellular fluid | 0.3 | L/min | |
| $V_{plasma}$ | plasma fluid volume | 4.5 | L | [29] |
| $V_{interstitial}$ | interstitial fluid volume | 10 | L | [29] |
| $V_{max}$ | maximum $K^+$ flow into cells | 130 | mEq/min | [30] |
| $K_m$ | half maximal activation | 1.4 | mEq/L | [30] |
| $P_{trans}$ | effective transmembrane $K^+$ permeability | 0.697 | L/min | |
| $V_{intracellular}$ | intracellular fluid volume | 24 | L | [29] |
| $\Phi_{GFR}$ | glomerular filtration rate | 0.125 | L/min | [29] |
| $\Phi^{eq}_{dt-Ksec}$ | baseline distal tubule $K^+$ secretion | 0.0410 | mEq/min | [7] |
| $\Phi^{eq}_{cd-Ksec}$ | baseline collecting duct $K^+$ secretion | 0.0022 | mEq/min | [7] |
| $A_{cd-Ksec}$ | parameter that determines $\lambda_{al}$ (Eq 24) | 0.161 | | |
| $B_{cd-Ksec}$ | parameter that determines $\lambda_{al}$ (Eq 24) | 0.411 | | |
| $A_{cd-Kreab}$ | parameter that determines $\Phi_{cd-Kreab}$ (Eq 17) | 0.499 | | |
| $T_{al}$ | ALD regulatory time constant | 60 | mins | [29] |
| $C_{sod}$ | plasma $Na^+$ concentration | 144 | mEq/L | [29] |
| $A_{dt-Ksec}$ | parameter that determines $\gamma_{al}$ (Eq 23) | 0.348 | | |
| $B_{dt-Ksec}$ | parameter that determines $\gamma_{al}$ (Eq 23) | 0.238 | | |
| $A_{insulin}$ | parameter that determines $\rho_{insulin}$ (Eq 26) | 0.999 | | |
| $B_{insulin}$ | parameter that determines $\rho_{insulin}$ (Eq 26) | 0.665 | | |
| $\alpha_{insulin}$ | parameter that determines $\rho_{insulin}$ (Eq 26) | 1.068 | | |
| $\beta_{insulin}$ | parameter that determines $\rho_{insulin}$ (Eq 26) | 0.538 | | |
| $A_{Kin}$ | parameter that determines $\gamma_{Kin}$ (Eq 27) | 0.250 | | |

Note that the total extracellular fluid is about 14.5 L in the average male, where about 4.5 L is plasma fluid [29]. Letting the volume of plasma be given by $V_{\text{plasma}}$ = 4.5 L and interstitial volume $V_{\text{inter}}$ = 10 L, we have

$$K_{\text{plasma}} = \frac{M_{\text{Kplasma}}}{V_{\text{plasma}}} \tag{4}$$

$$K_{\text{interstital}} = \frac{M_{\text{Kinter}}}{V_{\text{inter}}} \tag{5}$$

where $M_{\text{Kinter}}$ is the amount of K$^+$ in the interstitial fluid and is determined by

$$\frac{dM_{\text{Kinter}}}{dt} = \Phi_{\text{ECF}} - \Phi_{\text{ECtoIC}} + \Phi_{\text{ICtoEC}} \tag{6}$$

where $\Phi_{\text{ECtoIC}}$ and $\Phi_{\text{ICtoEC}}$ are the flux of K$^+$ from the extracellular to the intracellular fluid and flux of K$^+$ from the intracellular to the extracellular fluid, respectively (described below).

**Intracellular fluid.** Most K$^+$ (about 75%) within the intracellular fluid is stored in skeletal muscle cells [3]. Hence, the amount of K$^+$ in the skeletal muscle is the major driving factor in intracellular K$^+$ concentration. The amount of K$^+$ in the intracellular compartment is denoted by $M_{\text{KIC}}$ and is determined by

$$\frac{dM_{\text{KIC}}}{dt} = \Phi_{\text{ECtoIC}} - \Phi_{\text{ICtoEC}}. \tag{7}$$

Flow of K$^+$ into the intracellular fluid ($\Phi_{\text{ECtoIC}}$) is driven by Na$^+$-K$^+$-ATPase. This is modeled using Michaelis-Menten kinetics so that

$$\Phi_{\text{ECtoIC}} = \rho_{\text{al}} \times \rho_{\text{insulin}} \times \frac{V_{\text{max}} \times K_{\text{interstital}}}{K_{\text{m}} + K_{\text{interstitial}}} \tag{8}$$

where $V_{\text{max}}$ is the maximum rate, $K_{\text{m}}$ denotes the half maximal activation level, $\rho_{\text{al}}$ is the effect of ALD on Na$^+$-K$^+$-ATPase, and $\rho_{\text{insulin}}$ is the effect of insulin (see Eqs 22 and 26 below for more details).

Flow from intra- to extracellular compartments (i.e., $\Phi_{\text{ICtoEC}}$) is driven by diffusion through a permeable membrane

$$\Phi_{\text{ICtoEC}} = P_{\text{trans}} \times (K_{\text{IC}} - K_{\text{interstitial}}) \tag{9}$$

where $P_{\text{trans}}$ denotes the transmembrane permeability, $K_{\text{IC}}$ is the concentration of K$^+$ in the intracellular fluid. Note that

$$K_{\text{IC}} = \frac{M_{\text{KIC}}}{V_{\text{intracellular}}} \tag{10}$$

where $V_{\text{intracellular}}$ is the volume of the intracellular space (see Table 1).

## Renal handling of K$^+$

Potassium is freely filtered at the glomerulus. As such, the filtered K$^+$ load ($\Phi_{\text{filK}}$) is the glomerular filtration rate (GFR; $\Phi_{\text{GFR}}$) multiplied by the plasma K$^+$ concentration, i.e.,

$$\Phi_{\text{filK}} = \Phi_{\text{GFR}} \times K_{\text{plasma}}. \tag{11}$$

Note that, in this model, $\Phi_{\text{GFR}}$ is assumed to be known *a prior* based on average glomerular filtration rate (see Table 1).

As the filtrate passes through the nephrons, $K^+$ is reabsorbed and returned to the general circulation along some tubular segments, and secreted into the lumen along the distal segments. $K^+$ transport is coupled to other solutes, either directly via a cotransporter or indirectly via membrane potential. But for simplicity, the transport of other solutes and fluid is not represented in this model. Eventually, what is remaining at the end of the collecting duct is excreted in urine. Despite the heterogeneity among nephron populations, for simplicity the model represents a single nephron, divided into three model segments: the "proximal segment (ps)" which includes the proximal tubule and the loop of Henle, the "distal segment (dt)" that includes the distal convoluted tubule and the connecting tubule, and the collecting duct (cd).

The proximal segment is assumed to reabsorb a fixed fraction (92%) of the filtered $K^+$ based on Ref. [7]. Fractional reabsorption of $K^+$ in the proximal segment (proximal tubule and thick ascending limb) remains largely the same under low, normal, and high $K^+$ intake conditions [7]. Therefore, proximal segment $K^+$ reabsorption is modeled by

$$\Phi_{ps-Kreab} = 0.92 \times \Phi_{filK}. \tag{12}$$

The proximal segment (which includes the loop of Henle) ends at the macula densa. Macula densa $K^+$ flow ($\Phi_{mdK}$) is the difference between filtered $K^+$ load and proximal tubule reabsorption:

$$\Phi_{mdK} = \Phi_{filK} - \Phi_{ps-Kreab}. \tag{13}$$

The macula densa marks the beginning of the distal tubule, where most of the renal $K^+$ handling is finely regulated.

Along the distal convoluted tubule and connecting tubule (we refer to these segments together as the "distal tubule") $K^+$ secretion occurs. Distal tubule $K^+$ secretion ($\Phi_{dt-Ksec}$) has a baseline value of $\Phi_{dt-Ksec}^{eq}$ (see Table 1), chosen to yield a urine $K^+$ flow of $0.9 \times \Phi_{Kin}^{ss}$ at steady state. Distal tubule $K^+$ secretion is regulated by ALD and the $K^+$ intake-mediated feedforward mechanism so that

$$\Phi_{dt-Ksec} = \Phi_{dt-Ksec}^{eq} \times \gamma_{al} \times \gamma_{Kin} \tag{14}$$

where $\gamma_{al}$ denotes the regulatory effect of ALD and $\gamma_{Kin}$ represents the feedforward effect of $K^+$ intake (see Eqs 22 and 27 below for details).

Distal tubule $K^+$ outflow ($\Phi_{dtK}$; mEq/min) is given by the macula densa $K^+$ flow plus net distal tubule $K^+$ secretion:

$$\Phi_{dtK} = \Phi_{mdK} + \Phi_{dt-Ksec} \tag{15}$$

which then enters the collecting duct. The collecting duct both secretes and reabsorbs $K^+$ through the function of principal and intercalated cells, respectively [7]. Collecting duct $K^+$ secretion ($\Phi_{cd-Ksec}$) has a baseline value of $\Phi_{cd-Ksec}^{eq}$ (see Table 1), and is regulated by ALD. As such we let

$$\Phi_{cd-Ksec} = \Phi_{cd-Ksec}^{eq} \times \lambda_{al} \tag{16}$$

where $\lambda_{al}$ denotes the regulatory effect of ALD on collecting duct $K^+$ secretion (described below).

Collecting duct $K^+$ reabsorption ($\Phi_{cd-Kreab}$) is modeled as a fraction of distal tubule $K^+$ outflow:

$$\Phi_{cd-Kreab} = \Phi_{dtK} \times A_{cd-Kreab} \tag{17}$$

where $A_{cd-Kreab}$ is a fitting parameter (see Table 1).

Finally, urine K$^+$ excretion ($\Phi_{uK}$) is given by the difference between distal tubule K$^+$ outflow and net collecting duct K$^+$ transport ($\Phi_{cd-Ksec} - \Phi_{cd-Kreab}$) so that

$$\Phi_{uK} = \Phi_{dtK} + \Phi_{cd-Ksec} - \Phi_{cd-Kreab}. \tag{18}$$

## Aldosterone effects on cellullar K$^+$ uptake and kidney function

Aldosterone plays a major role in K$^+$ homeostasis via its effect on K$^+$ secretion along the distal tubule (see Eq 14) and Na$^+$-K$^+$-ATPase abundance (see Eq 8). To model ALD concentration ($C_{al}$), we let $N_{al}$ be the normalized ALD concentration so that

$$C_{al} = N_{al} \times 85 \ \text{ng/L} \tag{19}$$

where baseline $C_{al}$ is 85 ng/L. Then following the approach in Ref. [29] and letting $N_{als}$ denote normalized ALD secretion, we have that

$$\frac{dN_{al}}{dt} = \frac{1}{T_{al}} (N_{als} - N_{al}) \tag{20}$$

where $T_{al}$ is the time constant governing regulation of $C_{al}$.

In this model, ALD secretion is effected by letting

$$N_{als} = \ \max\left(0, \frac{K_{ECFtotal}/C_{sod}}{0.00972} - 2\right) \tag{21}$$

where $K_{ECFtotal} = (M_{Kplasma} + M_{Kinterstitial})/(V_{plasma} + V_{interstitial})$, $C_{sod}$ is the extracellular Na$^+$ concentration ($C_{sod}$ = 144 mEq/L), and is parameterized so that $N_{als}$ = 1 when $K_{ECFtotal}$ is at its steady state value (see Table 2).

An increase in ALD concentration leads to a modest increase in Na$^+$-K$^+$-ATPase abundance, which increases K$^+$ flux into cells [30–32]. We capture this effect by the scaling factor $\rho_{al}$ (from Eq 8), based on findings by Phakdeekitcharoen et al.[32]

$$\rho_{al} = \frac{66.4 \ \text{nmol/mg protein/h} + 0.273 \times C_{al}}{89.6050 \ \text{nmol/mg protein/h}}. \tag{22}$$

Distal tubule and collecting duct K$^+$ secretion both increase with increased ALD concentration. Field et al. [33] found that in adult male Sprague-Dawley rats, an increase in plasma ALD concentration from 4.4 ng/dl to 55.8 ng/dl caused distal tubule secretion to increase by 83%. Humans have a much higher baseline ALD concentration. Given that interspecies difference, we parameterize the equation so that the same percentage increase from the baseline ALD concentration induces the same percentage increase in secretion

$$\gamma_{al} = A_{dt-Ksec} \times (C_{al})^{B_{dt-Ksec}}. \tag{23}$$

where $A_{dt-Ksec}$ and $B_{dt-Ksec}$ are fitting parameters. Similarly, Schwartz et al. [34] measured K$^+$ flux in the cortical collecting ducts of rabbits at various ALD levels. Using this data, we fit the curve connecting ALD levels to collecting duct K$^+$ secretion so that

$$\lambda_{al} = A_{cd-Ksec} \times (C_{al})^{B_{cd-Ksec}} \tag{24}$$

where $A_{cd-Ksec}$ and $B_{cd-Ksec}$ are fitting parameters. See Table 1 for parameter values of $A_{dtK-sec}$, $B_{dtK-sec}$, $A_{cd-Ksec}$, and $B_{cd-Ksec}$.

**Table 2. Baseline model steady-state solution.** These values are used as the initial conditions in the model simulations.

| Variable | Description | Value | Unit |
|---|---|---|---|
| $M_{\text{Kgut}}$ | amount of K⁺ in gut and hepatoportal circulation | 4.4 | mEq |
| $M_{\text{Kplasma}}$ | amount of K⁺ in plasma | 18.9 | mEq |
| $M_{\text{Kinterstitial}}$ | amount of K⁺ in interstitial fluid | 42.1 | mEq |
| $M_{\text{KIC}}$ | amount of K⁺ in intracellular fluid | 3104.2 | mEq |
| $K_{\text{plasma}}$ | plasma K⁺ concentration | 4.2 | mEq/L |
| $K_{\text{interstitial}}$ | interstitial fluid K⁺ concentration | 4.2 | mEq/L |
| $K_{\text{IC}}$ | intracellular fluid K⁺ concentration | 129.3 | mEq/L |
| $\Phi_{\text{ECtoIC}}$ | K⁺ flow into cells | 97.6 | mEq/min |
| $\Phi_{\text{ICtoEC}}$ | K⁺ flow out of cells | 97.6 | mEq/min |
| $\Phi_{\text{filK}}$ | filtered K⁺ load | 0.53 | mEq/min |
| $\Phi_{\text{ps-Kreab}}$ | proximal segment K⁺ reabsorption | 0.48 | mEq/min |
| $\Phi_{\text{mdK}}$ | tubular K⁺ flow at macula densa | 0.042 | mEq/min |
| $\Phi_{\text{dt-Ksec}}$ | distal tubule K⁺ secretion | 0.041 | mEq/min |
| $\Phi_{\text{dtK}}$ | distal tubule K⁺ outflow | 0.083 | mEq/min |
| $\Phi_{\text{cd-Ksec}}$ | collecting duct K⁺ secretion | 0.0022 | mEq/min |
| $\Phi_{\text{cd-Kreab}}$ | collecting duct K⁺ reabsorption | 0.042 | mEq/min |
| $\Phi_{\text{uK}}$ | urine K⁺ flow | 0.044 | mEq/min |

## Insulin effect on cellular K⁺ uptake

We approximate the relationship between serum insulin levels and time after a meal based on Benedict et al. [35]. Letting $t_0$ denote the time at the beginning of the meal (in minutes) we have

$$
C_{\text{insulin}} = \begin{cases} \frac{(325-22.6)\ \text{pmol/L}}{90\ \text{min}} \times (t - t_0\ \text{min}) + 22.6\ \text{pmol/L} & \text{if } t_0 \leq t < t_0 + 90 \\[2mm] \frac{-(325-22.6)\text{pmol/L}}{270\ \text{min}} \times (t - (t_0 + 360\ \text{min})) + 22.6\ \text{pmol/L} & \text{if } t_0 + 90 \leq t \leq t_0 + 360 \\[2mm] 22.6\ \text{pmol/L} & \text{otherwise.} \end{cases} \tag{25}
$$

where $C_{\text{insulin}}$ denotes the concentration of serum insulin (pmol/L).

Serum insulin concentration stimulates Na⁺-K⁺-ATPase uptake [1, 36]. Thus, we let $\rho_{\text{insulin}}$ (from Eq 8) be given by

$$
\rho_{\text{insulin}} = \max\left(1, \frac{A_{\text{insulin}}}{1 + e^{-\alpha_{\text{insulin}}(\log(C_{\text{insulin}}/1000) - \log(\beta_{\text{insulin}}))}} + B_{\text{insulin}}\right) \tag{26}
$$

where $\alpha_{\text{insulin}}$, $A_{\text{insulin}}$, $\beta_{\text{insulin}}$, and $B_{\text{insulin}}$ are fitting parameters (see Table 1).

## Gastrointestinal feedforward control mechanism

In experimental animals, when plasma K⁺ concentration is artificially maintained constant, the kaliuretic response to a K⁺ load is greater when given as a meal than with an intravenous infusion [19, 37]. This evidence suggests the existence of K⁺ sensors in the gut capable of responding to dietary K⁺ to signal to the kidney to rapidly alter K⁺ excretion before plasma K⁺ levels increase. The model represents this feedforward effect via the term $\gamma_{\text{Kin}}$, which alters distal tubule K⁺ secretion with $M_{\text{Kgut}}$ (see Eq 14) where

$$
\gamma_{\text{Kin}} = \max\left(1, A_{\text{Kin}} \times (M_{\text{Kgut}} - M_{\text{Kgut}}^{ss}) + 1\right) \tag{27}
$$

where $A_{\text{Kin}}$ is a fitting parameter (see Table 1) and $M_{\text{Kgut}}^{ss}$ is the steady state value of $M_{\text{Kgut}}$. Note that we have

$$M_{\text{Kgut}}^{ss} = \frac{0.9 \times \Phi_{\text{Kin}}^{ss}}{k_{\text{gut}}}$$

by solving Eq 1 at steady state.

## Muscle-kidney cross talk mechanism

As previously noted, following a K$^+$ load or depletion, K$^+$ moves into or out of muscle cells as needed to stabilize plasma K$^+$ concentration. However, sustained high or low intracellular K$^+$ concentrations can be harmful. It has been hypothesized that a muscle-kidney cross talk signal may allow intracellular K$^+$ concentration $K_{\text{IC}}$ to more rapidly return to its baseline level, without significant variations in plasma K$^+$ concentration [1, 14, 19]. According to this hypothesis, a sufficiently high (muscle) $K_{\text{IC}}$ elevates urine K$^+$ excretion and vice versa.

Because there has not been definitive evidence for this muscle-kidney cross talk, it is not included in the baseline model. However, we conducted simulations for these "*what-if*" scenarios to study its potential impact. Muscle-kidney cross talk is represented in these scenarios via a coupling function that links intracellular K$^+$ to either (i) distal tubule K$^+$ secretion, (ii) collecting duct K$^+$ secretion, or (iii) collecting duct K$^+$ reabsorption. The coupling function ($\omega_{\text{Kic}}$) is taken to be a linearly increasing function of $K_{\text{IC}}$, bounded above 0:

$$\omega_{\text{Kic}} = \max\left(0, m_{\text{Kic}}(K_{\text{IC}} - K_{\text{IC}}^{\text{baseline}}) + 1\right) \tag{28}$$

where the value for $m_{\text{Kic}}$ depends on the case being simulated (see Eqs 29, 30, and 31).

At baseline, $K_{\text{IC}} = K_{\text{IC}}^{\text{baseline}} = 130\,\text{mEq/L}$, thus $\omega_{\text{Kic}} = 1$. To model a muscle-kidney cross talk signal that targets distal tubule K$^+$ secretion, $\omega_{\text{Kic}}$ can be added to Eq 14 so that

$$\Phi_{\text{dt}-\text{Ksec}} = \Phi_{\text{dt}-\text{Ksec}}^{eq} \times \gamma_{\text{al}} \times \gamma_{\text{Kin}} \times \omega_{\text{Kic}}, \qquad m_{\text{Kic}} = 0.1 \tag{29}$$

We will refer to this scenario as Case MKX-DT-sec. If collecting duct secretion is the target, then we modify Eq 16 by

$$\Phi_{\text{cd}-\text{Ksec}} = \Phi_{\text{cd}-\text{Ksec}}^{eq} \times \lambda_{\text{al}} \times \omega_{\text{Kic}}, \qquad m_{\text{Kic}} = 0.1. \tag{30}$$

Similarly, if collecting duct reabsorption is the target, we modify Eq 17 by

$$\Phi_{\text{cd}-\text{Kreab}} = \Phi_{\text{dtK}} \times A_{\text{cd}-\text{Kreab}} \times \omega_{\text{Kic}}, \qquad m_{\text{Kic}} = -0.1. \tag{31}$$

We refer to these two cases as Case MKX-CD-sec and Case MKX-CD-reab, respectively.

## Model parameters

Model parameter values are listed in Table 1. References are given for parameter values that were found in the literature; other parameters were fitted during model construction.

To fit $P_{\text{trans}}$, we applied the constraint that at steady state, the flows into and out of cells are equal. That is we set the right sides of Eqs 8 and 9 equal with the feedback effects set to 1 (due to steady state) and used the baseline values for $K_{\text{IC}}$ and $K_{\text{interstitial}}$ (130 mEq/L and 4.3 mEq/L, respectively), and solved for $P_{\text{trans}}$.

Phakdeekitcharoen et al. [32] found a linear relationship between aldosterone levels and skeletal muscle Na$^+$-K$^+$-ATPase activity. We fit Eq 22 to the results presented in their study normalized to baseline Na$^+$-K$^+$-ATPase activity to represent the impact of aldosterone concentration on K$^+$ uptake into the intracellular fluid. Field et al. [33] found that in adult male

Sprague-Dawley rats, an increase in plasma ALD concentration from 4.4 ng/dl to 55.8 ng/dl causes distal tubule K$^+$ secretion to increase by 83%. Our (human male) model has a much higher baseline ALD concentration ($C_{al}^{baseline} = 85$ ng/l). Given that discrepancy, we chose values for parameters $A_{dt-Ksec}$ and $B_{dt-Ksec}$ in Eq 23 so that the same percent increase from ALD baseline concentration causes the same percent increase in K$^+$ secretion. Schartz et al. [34] measured K$^+$ flux in the cortical collecting ducts of rabbits at various plasma ALD concentration. Using this data, we fit the curve connecting ALD levels to collecting duct K$^+$ secretion to get values for the parameters $A_{cd-Ksec}$ and $B_{cd-Ksec}$ in Eq 24.

Some model parameters had to be estimated by fitting to K$^+$ regulation data for a single meal intake. Specifically, the magnitude of the gastrointestinal feedforward effect ($A_{Kin}$), collecting duct K$^+$ reabsorption parameter ($A_{cd-Kreab}$), and insulin sensitivity parameters ($A_{insulin}$, $B_{insulin}$) were fitted to match model behavior to the plasma K$^+$ concentration and urine K$^+$ excretion data from Preston et al. [17].

## Numerical methods

Model simulations were conducted using a variable-step, variable-order implicit differential algebraic equation solver (`ode15i, MATLAB R2021a`). Parameter fitting was conducted using the interior-point algorithm (`fmincon, MATLAB R2021a`) to minimize model predicted values from given data. The code used for this study is available at https://github.com/Layton-Lab/Kregulation.

## Results

### Baseline steady-state results

We computed the baseline steady-state solution by assuming a daily K$^+$ intake of 70 mEq a day distributed evenly over 24 hours, corresponding to steady state K$^+$ intake of $\Phi_{Kin}^{ss} = 0.049$ mEq/min. We chose this value for $\Phi_{Kin}^{ss}$ due to the experimental procedures used in Preston et al. [17] that was used to fit the baseline model parameters. We also note that while the daily recommended intake of K$^+$ is 120 mEq/day, the average person on a Western diet consumes much less [19]. Using the model parameter values listed in Table 1, we integrated baseline model equations (Eqs 1–27) to steady state. Model steady state solution values are summarized in Table 2. We report normal ranges for K$^+$ amount and concentrations in Table 3.

At steady state, 97.9% of the total body K$^+$ is stored intracellularly. This is consistent with estimates in the literature [3, 28]. Model steady state plasma and intracellular K$^+$ concentrations are 4.2 and 129.3 mEq/L, respectively. These concentrations are within normal range [3]. Urine K$^+$ excretion is 0.044 mEq/min, which is 90% of $\Phi_{Kin}^{ss}$ reflecting that 90% of K$^+$ intake is excreted through urine, while the remaining 10% is excreted via feces [3]. The filtered K$^+$ load ($\Phi_{filK}^{ss} = 0.53$ mEq/min) is determined by the fixed GFR parameter ($\Phi_{GFR}$) and the steady-state plasma K$^+$ concentration ($K_{plasma}^{ss}$). About 92% of the filtered K$^+$ is reabsorbed along the

**Table 3. Normal ranges for K$^+$ concentration and amount.**

| Compartment | Range | Unit | Ref |
|---|---|---|---|
| Amount of K$^+$ in extracellular fluid | 60–70 | mEq | [1, 3] |
| Amount of K$^+$ in intracellular fluid | 3400—3900 | mEq | [3] |
| Extracellular K$^+$ concentration | 3.5–5.0 | mEq/L | [1, 3] |
| Intracellular K$^+$ concentration | 120–140 | mEq/L | [1, 3] |

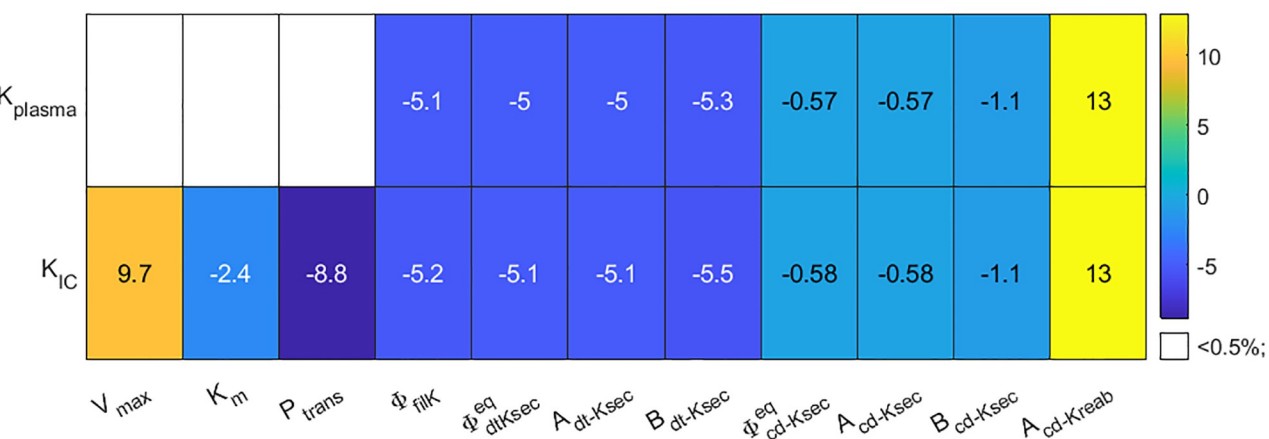

**Fig 3. Local sensitivity analysis.** Percent changes in baseline model steady state solution for plasma K⁺ concentration ($K_{plasma}$) and intracellular K⁺ concentration ($K_{IC}$) in response to a perturbation of 10% increase of the given parameter value.

proximal segments, followed by substantial K⁺ secretion along the distal tubule ($\Phi^{ss}_{dt-Ksec}$ = 0.041 mEq/min). Both reabsorption and secretion happens along the collecting duct, resulting in net reabsorption at steady state ($\Phi^{ss}_{cd-Kreab} - \Phi^{ss}_{cd-Ksec}$ = 0.039 mEq/min). Taken together, urine K⁺ excretion corresponds to about 8.3% of the filtered K⁺ load.

**Sensitivity analysis.** We conducted a local sensitivity analysis by increasing selected parameters by 10% and computing the resulting changes in steady state plasma and intracellular K⁺ concentration. Results are shown in Fig 3.

Among the parameters that affect renal handling of K⁺ (i.e., filtration, distal tubule K⁺ secretion, and collecting duct K⁺ secretion and reabsorption), plasma and intracellular K⁺ concentration are most sensitive to $A_{cd-Kreab}$ (Fig 3). This sensitivity may, in large part, attributed to the lack of downstream segments that can compensate changes in collecting duct K⁺ reabsorption. As such increasing $A_{cd-Kreab}$ by 10% increases collecting duct K⁺ reabsorption proportionally (see Eq 17), significantly lowering K⁺ excretion, which leads to a 13% increase in plasma and intracellullar K⁺ concentrations (Fig 3).

Intracellular K⁺ concentration is also highly sensitive to parameters that impact the K⁺ fluxes in and out of the intracellular compartment (Fig 3): the Michaelis-Menten parameter $V_{max}$ (Eq 8) and transmembrane permeability $P_{trans}$ (Eq 9). In contrast, plasma K⁺ concentration is insensitivite to changes in these parameters, as explained below. Recall that in this analysis K⁺ intake is fixed, so then an increase in $V_{max}$ will cause an immediate influx of K⁺ into the intracellular compartment, which decreases plasma K⁺. This causes a proportional decrease in K⁺ nephron filtration, which in turn decreases urinary K⁺ excretion. Together, the fixed K⁺ intake and (transiently) lower K⁺ excretion yield net K⁺ retention, which accumulates in the extracellullar K⁺ compartments until plasma K⁺ concentration is back to baseline, with urinary K⁺ matching K⁺ intake, and a new steady state is obtained.

## Response to a single meal of varying content

What are the physiological responses to meals with different contents? We followed experimental protocols in Preston et al. [17] and simulated a K⁺ deficient meal (referred to as "Meal only" experiment), 35 mEq of orally ingested K⁺ only (referred to as "KCl only" experiment), and a K⁺ deficient meal with 35 mEq of K⁺ (referred to as the "Meal + KCl" experiment). This

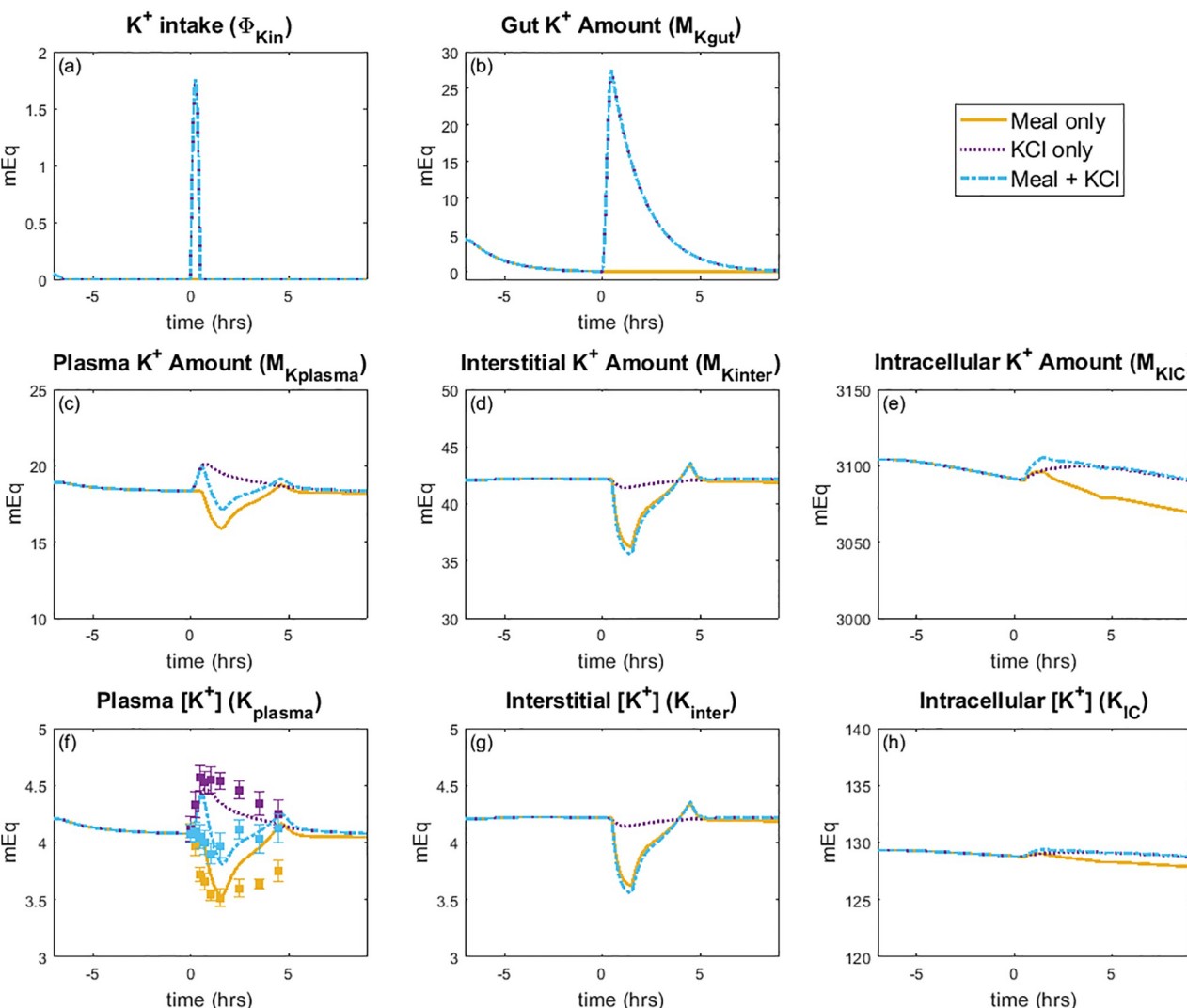

**Fig 4. Extracellular and intracellular K⁺ response to a single meal.** *(A)* Prescribed K⁺ intake, *(B)-(E)* predicted K⁺ amount in various compartments, and *(F)-(H)* predicted K⁺ concentrations for a single meal load. The meal experiment occurs at time = 0 hours. Initial conditions are the baseline steady state values (see Table 2) followed by a fasting state set before the meal experiment. The yellow lines denote the K⁺ deficient meal (i.e., "Meal only" experiment); purple, only K⁺ ingestion (i.e., "KCl only" experiment); blue, meal with K⁺ (i.e., "Meal + KCl" experiment). Points and error bars for plasma K⁺ concentration *(f)* show experimental data from Preston et al. [17].

was accomplished by varying K⁺ intake ($\Phi_{Kin}$) as an independent variable. Initial conditions were taken to be the baseline steady state solution (Table 2). To simulate a fasting state, K⁺ intake was initialized at the steady state value ($\Phi_{Kin}^{ss} = 0.049$ mEq/min), then decreased to $\Phi_{Kin} = 0$ mEq/min and kept at no K⁺ intake for six hours (see Fig 4A). After this fasting period, a meal was simulated depending on the experiment type (i.e., "Meal only", "KCl only", or "Meal + KCl"). For the experiments that include K⁺ ingestion (i.e., "KCl only" and "Meal + KCl"), a dose of 35 mEq of K⁺ is given over 30 minutes of meal ingestion (see Fig 4A). Fig 4 shows simulation results for the impact of K⁺ intake on K⁺ amounts and concentrations in the gut, plasma, interstitial, and intracellular fluid for each of the three experiments. Fig 5 shows the effects of K⁺ intake on the kidney and Fig 6 shows the predicted effects of insulin and ALD as well as the gastrointestinal feedforward effect.

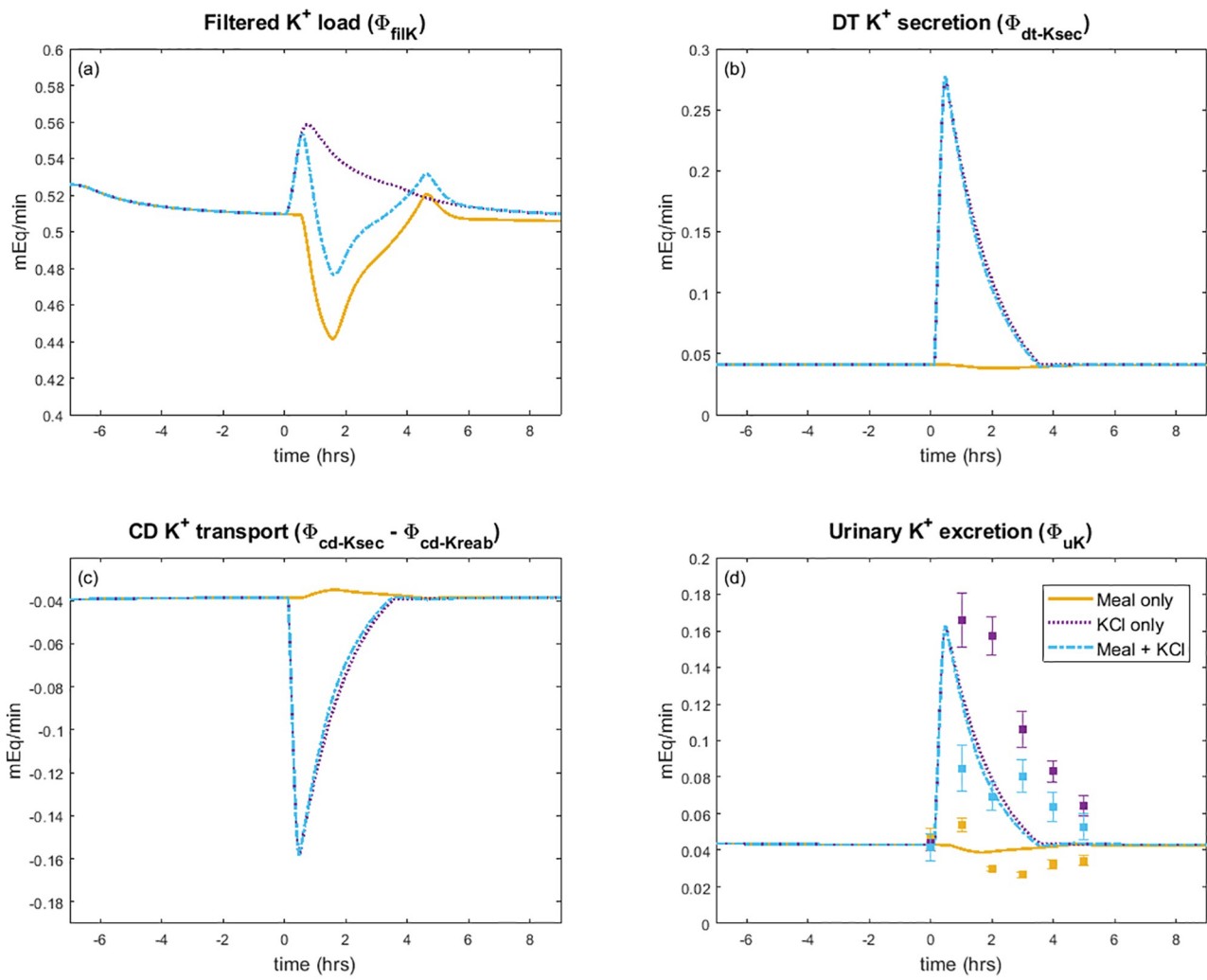

**Fig 5. Renal K$^+$ response to a single meal.** Model simulation predictions for kidney K$^+$ handling during a single meal load. The meal experiment occurs at time = 0 hours. Initial conditions are the baseline steady state values (see Table 2) with a fasting state before the meal experiment. The yellow lines denote the K$^+$ deficient meal (i.e., "Meal only" experiment); purple, only K$^+$ ingestion (i.e., "KCl only" experiment); blue, meal with K$^+$ (i.e., "Meal + KCl" experiment). Points and error bars for urine K$^+$ excretion show experimental data from Preston et al. [17].

When a meal containing glucose is ingested, insulin is secreted. Besides its role in glucose metabolism, insulin also activates cellular uptake of K$^+$ by increasing Na$^+$-K$^+$-ATPase activity. To isolate insulin's effect on K$^+$ regulation, we simulate a K$^+$ deficient meal ("Meal only" experiment) by running the baseline model with $\Phi_{Kin} = 0$, with the stimulation of insulin as given by Eqs 25 and 26. The model predicts that the insulin in a K$^+$-deficient meal elevates Na$^+$-K$^+$-ATPase uptake, by up to 10% (Fig 6A), resulting in an immediate increase of about 5 mEq in the intracellular K$^+$ fluid (Fig 4E). However, that increase is small relative to the base-line intracellular K$^+$ amount (3104.2 mEq; Table 2); thus, there is essentially no change in the intracellular K$^+$ concentration (Fig 4H). In contrast, the baseline amount of K$^+$ in plasma and interstitial fluids are significantly less than in the intracellular fluid (18.9 and 42.1 mEq, respectively, vs. 3104.2 mEq; Table 2); thus, the higher cellular K$^+$ uptake is enough to reduce both the interstitial and plasma K$^+$ concentration from about 4.0 mEq/L at the end of the fasting

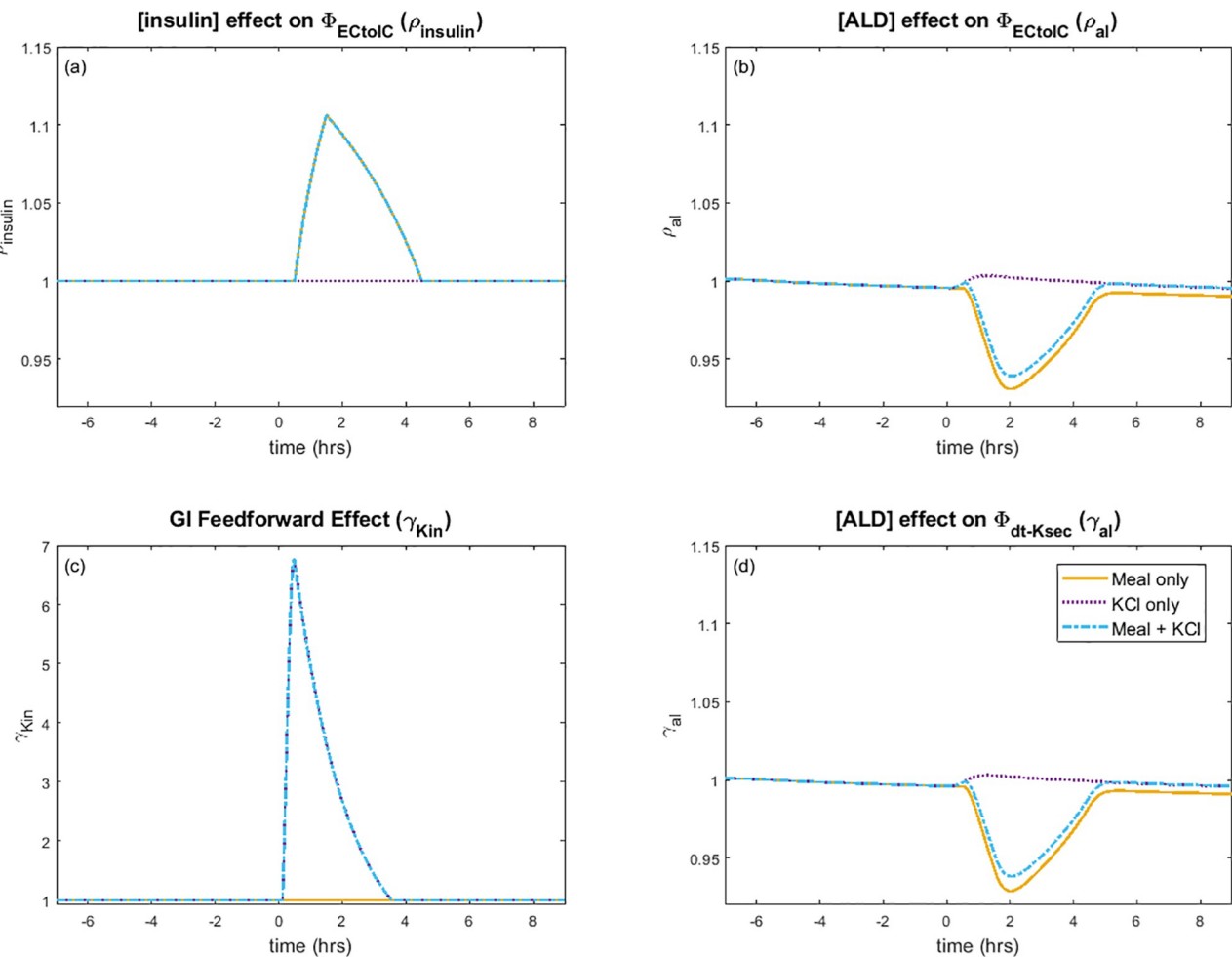

**Fig 6. Feedforward and feedback response to a single meal.** Model simulation predictions for *(A)* effect of insulin concentration ([insulin]) on $Na^+$-$K^+$-ATPase $K^+$ uptake, *(B)* aldosterone concentration effect on $Na^+$-$K^+$-ATPase $K^+$ uptake, *(C)* gastrointestinal (GI) feedforward effect on distal tubule $K^+$ secretion, and *(D)* aldosterone effect on distal tubule $K^+$ secretion. The meal experiment occurs at 0 hours. Initial conditions are steady state with a fasting state set before the meal experiment. The yellow lines denote the $K^+$ deficient meal (i.e., "Meal only" experiment); purple, only $K^+$ ingestion (i.e., "KCl only" experiment); blue, meal with $K^+$ (i.e., "Meal + KCl" experiment).

period to about 3.5 mEq/L (Fig 4F; Fig 4G). The predicted trend in plasma $K^+$ concentration is consistent with data reported by Preston et al. [17] (Fig 4F). This simulation illustrates the mechanism by which insulin lowers the extracellular $K^+$ concentration without significantly elevating intracellular $K^+$ concentration following a meal. We also note that the predicted urine $K^+$ excretion (Fig 5D) captures the observed trend of a decrease in urine $K^+$ excretion [17].

Next we considered a $K^+$ load not accompanied by a meal containing glucose ("KCl only"). We did this by increasing the independent variable $\Phi_{Kin}$ so that there is a 35 mEq dose of $K^+$ (Fig 4A), but did not include the effect of insulin (i.e., $\rho_{insulin} = 1$; Fig 6A). When $K^+$ is added into the model system, the gut $K^+$ amount increases from 0 mEq to a peak of 27 mEq (Fig 4B). The $K^+$ is eventually absorbed into the plasma, causing plasma $K^+$ concentration to increase by 12% (Fig 4F). The gastrointestinal feedforward effect from the increased gut $K^+$ amount (Fig 6C) signals to the distal tubule to rapidly increase $K^+$ secretion (Fig 5B). This in turn increases

 

urine $K^+$ excretion so that plasma $K^+$ levels can return to baseline after about 6 hours (Fig 4F). The predicted plasma $K^+$ concentration and urine $K^+$ excretion (Figs 4F and 5D) match the observed trends [17].

The baseline model represents a typical meal that contains both $K^+$ and glucose, therefore includes both the effect of insulin on cellular uptake and gastrointestinal feedforward effect of $K^+$. The model predicts an initial rise in plasma $K^+$ concentration, but then the insulin effect on cellular uptake ($\rho_{insulin}$) leads to about 14 mEq $K^+$ being moved into the intracellular compartment, quickly returning the plasma $K^+$ concentration back to baseline level (Fig 4C). As in the "Meal only" case, there is no noticeable change in intracellular $K^+$ concentration (Fig 4H). The gastrointestinal feedfoward effect elevates distal tubule $K^+$ secretion, which in turn increases urine $K^+$ excretion (Figs 5 and 6). Together these effects stabilize plasma $K^+$ concentration around its baseline value.

## Effect of feedforward and feedback controls

We assessed the effect of the feedforward and feedback controls on $K^+$ regulation by conducting *in silico* experiments of the baseline model without individual control mechanisms. In these simulations, three equal typical meals were given (i.e., including glucose) each day, totaling 100 mEq of $K^+$ intake per day for 10 days (see Fig 7A).

With all control mechanisms intact, the baseline model maintains a relatively steady plasma $K^+$ concentration through the day with fluctuations around meal times (Fig 7B). Each day the predicted plasma $K^+$ concentration varied between 3.8 and 4.4 mEq/L with an average daily plasma $K^+$ concentration of 4.2 mEq/L (Fig 7B); intracellular $K^+$ concentration remained stable at about 130 mEq/L throughout the 10 day simulation (Fig 7C). Both plasma $K^+$ and intracellular $K^+$ concentration were within normal ranges for the entire 10 day intake simulation.

Both insulin and aldosterone affect $K^+$ balance by stimulating uptake of $K^+$ into the intracellular space via the $Na^+$-$K^+$-ATPase pump. When the effect of insulin on $Na^+$-$K^+$-ATPase uptake is turned off (i.e., $\rho_{insulin} = 1$), less $K^+$ enters the intracellular fluid, with plasma $K^+$ concentration reaching an average low of about 4.1 mEq/L, compared to the baseline value of 3.8 mEq/L. When the effect of ALD is turned off (i.e., $\rho_{al} = 1$), at meal times, predicted plasma $K^+$ concentration increases similarly to the baseline model. But as plasma $K^+$ begins to decrease after the meal, without the inhibitory effect of the resulting low ALD on $Na^+$-$K^+$-ATPase activity, plasma $K^+$ concentration dips below normal range to 3.4 mEq/L (Fig 7B). Over time, it seems that through the day plasma $K^+$ concentration is able to stabilize and intracellular $K^+$ concentration is slightly higher at the end of the 10 days at 132 mEq/L. We also simulated the effect of turning off $\gamma_{al}$ and $\lambda_{al}$, which affect distal tubule and collecting duct $K^+$ secretion, respectively, but they did not have a notable predicted impact on plasma or intracellular $K^+$ concentration (not shown in Fig 7).

The gastrointestinal feedforward effect is predicted to have the largest impact on $K^+$ regulation. Specifically, when this effect is turned off (i.e., $\gamma_{Kin} = 1$), on day 1 of normal $K^+$ intake the peak plasma $K^+$ concentration is 7% higher than the baseline model (Fig 7B). Each day the average plasma $K^+$ concentration and intracellular $K^+$ concentration increase so that at the end of the 10 days, average intracellular and plasma $K^+$ concentration are 9% higher than day 1 (Fig 7B and 7C). This shows that without the gastrointestinal feedfoward effect, total body $K^+$ increases by the same percentage in each of the compartments, hence increasing both plasma and intracellular $K^+$ concentrations. With continued normal $K^+$ intake, intracellular $K^+$ concentration is predicted to rise higher than normal range.

When all the feedforward and feedback effects are turned off, there is a large impact on both plasma and intracellular $K^+$ concentration (Fig 7B and 7C). Both plasma $K^+$ and

 

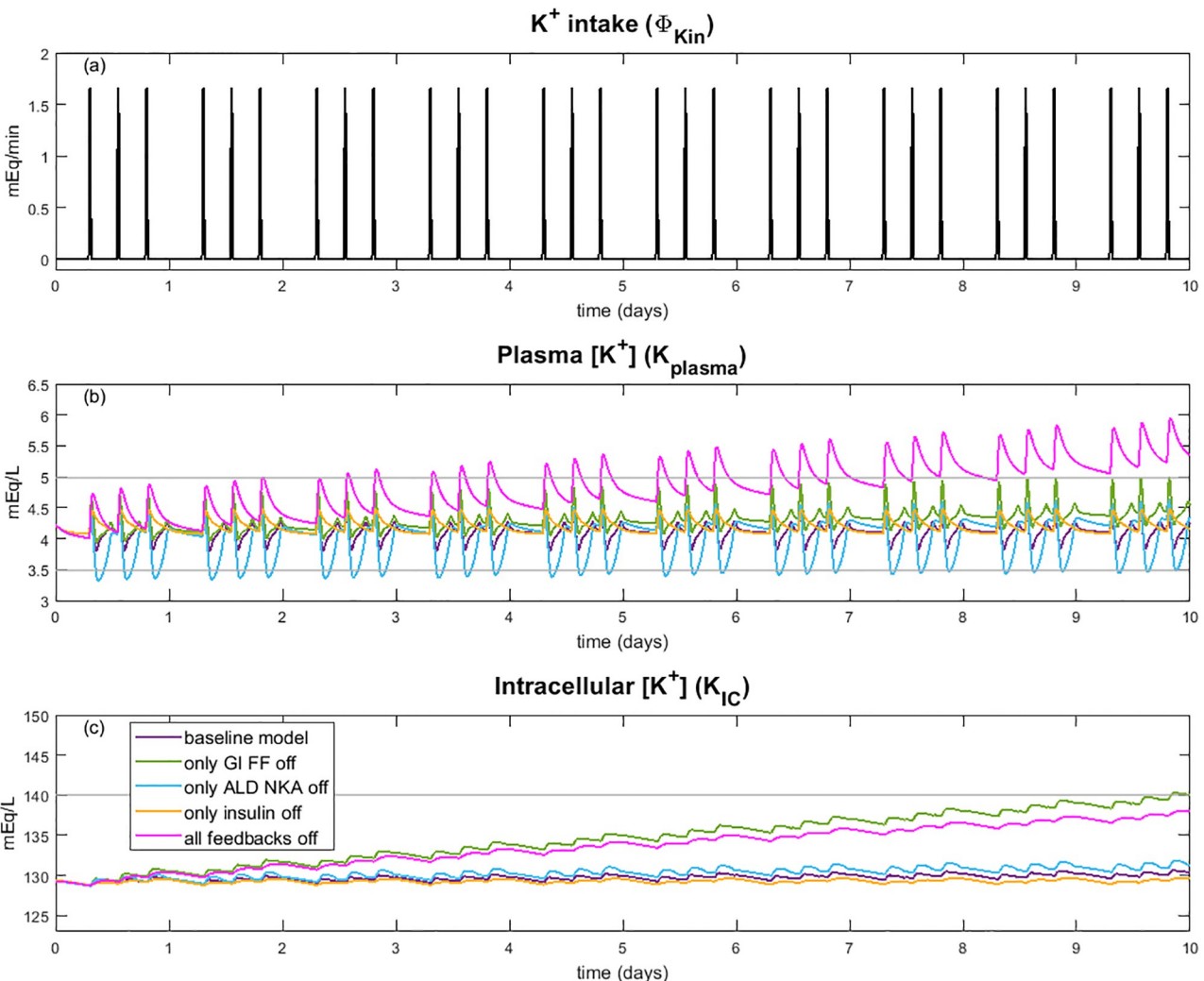

**Fig 7. Effect of feedforward and feedback controls.** Simulation results for plasma K$^+$ concentration *(B)* and intracellular K$^+$ concentration *(C)* for 10 days of normal K$^+$ intake (*(A)*;100 mEq/day, 3 typical meals) for the baseline model, baseline without gastrointestinal feedforward (GI FF) effect ($\gamma_{Kin}$ = 1), baseline without the ALD effect on Na$^+$-K$^+$-ATPase uptake ($\rho_{al}$ = 1), baseline model without insulin effect on Na$^+$-K$^+$-ATPase update ($\rho_{insulin}$ = 1), and baseline model with all feedback effects off. Grey lines indicate normal range for $K_{plasma}$ and $K_{IC}$.

intracellular K$^+$ concentrations increase so that by day 6, average plasma K$^+$ concentration is above normal range (Fig 7B). At the end of the 10 day simulation, peak plasma K$^+$ concentration is a severe hyperkalemic level of 5.9 mEq/L. Notably, while intracellular K$^+$ concentration does steadily increase through the 10 day simulation, the increase is not as high as when only the gastrointestinal feedforward effect is turned off. This is likely because insulin and aldosterone (eliminated in this case) both stimulate K$^+$ uptake into the intracellular space. Without this stimulation, the muscle cells fail to sufficiently buffer the increase in plasma K$^+$ concentration, which reaches a dangerously high level. Notably, average daily plasma K$^+$ concentration increases by 33% while intracellular K$^+$ concentration increases by 7%, showing that it is not only whole body K$^+$ that is increasing, but the distribution of K$^+$ between the extra- and intracellular fluid that is affected by not having all of these signals activated.

### Response to repeated K⁺ loading

How does the model respond to repeated K⁺ loading, and what might the effect of muscle-kidney cross talk be? To answer this question, we initialized the model at steady state and then for 2 days inputted 3 meals of equal amounts so that total K⁺ consumption was 100 mEq per day. Then for the next 4 days we increased the meal K⁺ amounts so that the total K⁺ consumption was 400 mEq per day. After the 4 days of K⁺ loading, we returned to the baseline of 100 mEq per day. The 14-day K⁺ intake values are shown in Fig 8A. This protocol is similar to the experiments conducted in Rabelink et al. [38]. Simulations were conducted for the baseline model, and separately with each of the 3 muscle-kidney cross talk mechanisms (Case MKX-DT-sec, Case MKX-CD-sec, and Case MKX-CD-reab).

When muscle-kidney cross talk is implemented to target collecting duct K⁺ secretion (Case MKX-CD-sec), the model predicts negligible effects (results not included in Fig 8). This is likely because baseline collecting duct K⁺ secretion is much smaller than either collecting duct K⁺ reabsorption or distal tubule K⁺ secretion (see Table 2). Muscle-kidney cross talk mechanisms that target distal tubule K⁺ secretion or collecting duct K⁺ reabsorption yield significant and similar effects on plasma K⁺ concentration and intracellular K⁺ concentration (Fig 8).

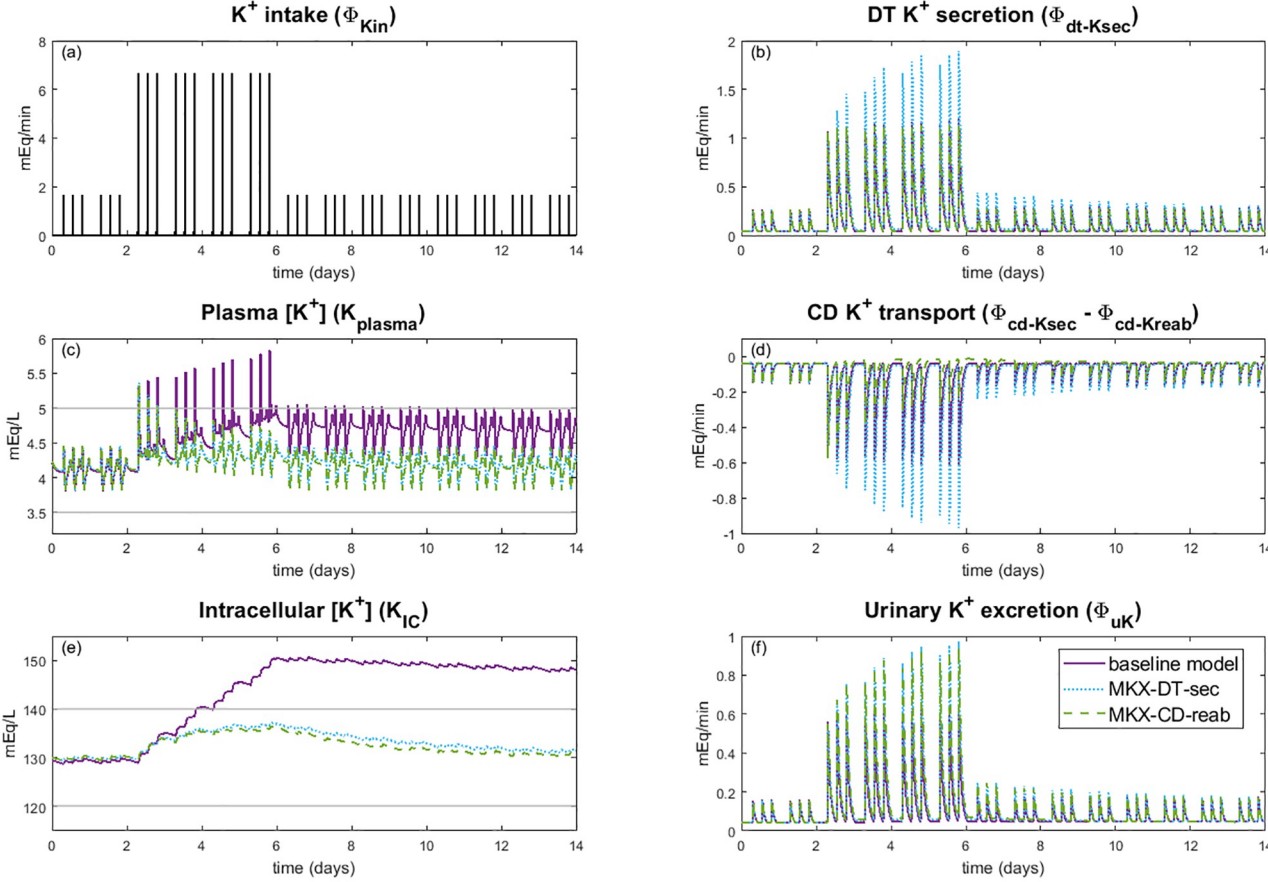

**Fig 8. Impact of muscle-kidney cross talk on K⁺ loading.** Simulation results for K⁺ loading experiments for the baseline model and muscle-kidney cross talk simulations (Case MKX-DT-sec and Case MKX-CD-reab). Potassium intake *(A)* is the same for all three simulation types. Note that Case MKX-CD-sec is not plotted since it had little impact from the baseline model results. Horizontal grey line shows normal range for plasma K⁺ concentration *(C)* and intracellular K⁺ concentration *(E)*. CD: collecting duct, DT: distal tubule

The baseline model, which does not have any muscle-kidney cross talk signal, predicts a strong response to a high $K^+$ diet (days 3–6). During high $K^+$ intake, the average total body $K^+$ increases by 16.2%, with most of this excess $K^+$ stored in the intracellular compartment. With the inclusion of the distal tubule muscle-kidney cross talk mechanism (Case MKX-DT-sec), average distal tubule $K^+$ secretion increases to 57.5% above baseline. For Case MKX-CD-reab, collecting duct reabsorption is inhibited so that on day 6, average collecting duct $K^+$ reabsorption is 61.6% below baseline. The two muscle-kidney cross talk mechanisms have a similar impact on urine $K^+$ excretion; on day 6, average urine $K^+$ excretion is ∼47% above baseline (Fig 8F). As a result, at the end of day 6, the peak intracellular $K^+$ concentration is about 137 mEq/L with muscle-kidney cross talk compared to 150 mEq/L for the baseline model. At the end of the 14 days, intracellular $K^+$ concentration remains out of normal range for the baseline model at 148.5 mEq/L, whereas intracellular $K^+$ concentration is near baseline values at 131 mEq/L with muscle-kidney cross talk (Fig 8E). Similarly, mean plasma $K^+$ concentration at day 6 is about 4.3 mEq/L and 4.9 mEq/L, respectively, with and without muscle-kidney cross talk (Fig 8C). While the latter is within normal range *on average*, it is on the high end and its peak value easily went out of range to a maximum of 5.8 mEq/L, which would be diagnosed as hyperkalemia. Indeed, repeated $K^+$ loading results in an increasing trend in the plasma $K^+$ concentration in the baseline model, but with muscle-kidney cross talk (Case MKX-DT-sec and Case MKX-CD-reab) plasma $K^+$ concentrations stays within a tight range. In other words, by signalling to the kidneys to excrete more $K^+$ when there are high intracellular $K^+$ concentrations, the body is better able to handle significant $K^+$ loads, thus preventing hyperkalemia. Model predictions suggest that the signal to collecting duct $K^+$ reabsorption or to distal tubule $K^+$ secretion would have a similar effect.

## Response to $K^+$ depletion

We next analyzed the model's response to $K^+$ depletion. To that end, we started by simulating a normal daily $K^+$ intake of 100 mEq per day. Then we restricted the model's daily $K^+$ intake to 25 mEq a day for 30 days to see when or if the model would predict hypokalemia. After the 30 day depletion period we returned $K^+$ intake to 100 mEq per day (Fig 9A). Simulations were conducted with and without each of the 3 muscle-kidney cross talk mechanisms (Case MKX-DT-sec, Case MKX-CD-sec, and Case MKX-CD-reab). Similar to the $K^+$ loading experiment done previously (Fig 8), Case MKX-CD-sec had little effect on the model simulation results from baseline (not shown).

During $K^+$ depletion, total body $K^+$ rapidly declines by 27.5% in the absence of muscle-kidney cross talk, with intracellular $K^+$ concentration reduced from ∼130 mEq/L to 93.8 mEq/L, significantly below normal range of 120–140 mEq/L (Fig 9E). Plasma $K^+$ concentration decreases from an average of 4.1 mEq/L to a severely hypokalemic daily average plasma $K^+$ concentration of 2.9 mEq/L by day 30 of $K^+$ depletion period (Fig 9C; baseline model).

Model simulations predict that muscle-kidney cross talk can be highly protective in retaining $K^+$ by massively decreasing urine $K^+$ as intracellular $K^+$ decreases, thereby preserving total body $K^+$ and plasma $K^+$. Specifically, Case MKX-DT-sec reduces distal tubule $K^+$ secretion to almost 0, resulting in a 30.8% decrease in average urine $K^+$ from the baseline model on day 30 of $K^+$ depletion (Fig 9B; Fig 9F). Case MKX-CD-reab more than doubles collecting duct $K^+$ reabsorption so that average urine $K^+$ is 46.0% lower than baseline on day 30 of $K^+$ depletion (Fig 9D and 9F). These drastically reduced urine $K^+$ excretion rates limit $K^+$ loss, resulting in only 11.5% and 5.3% reduction in total $K^+$ for Case MKX-DT-sec and Case MKX-CD-reab, respectively, compared to 27.5% in baseline. Consequently, intracellular and plasma $K^+$ concentrations remain within normal ranges. Notably, in Case MKX-CD-reab neither plasma nor

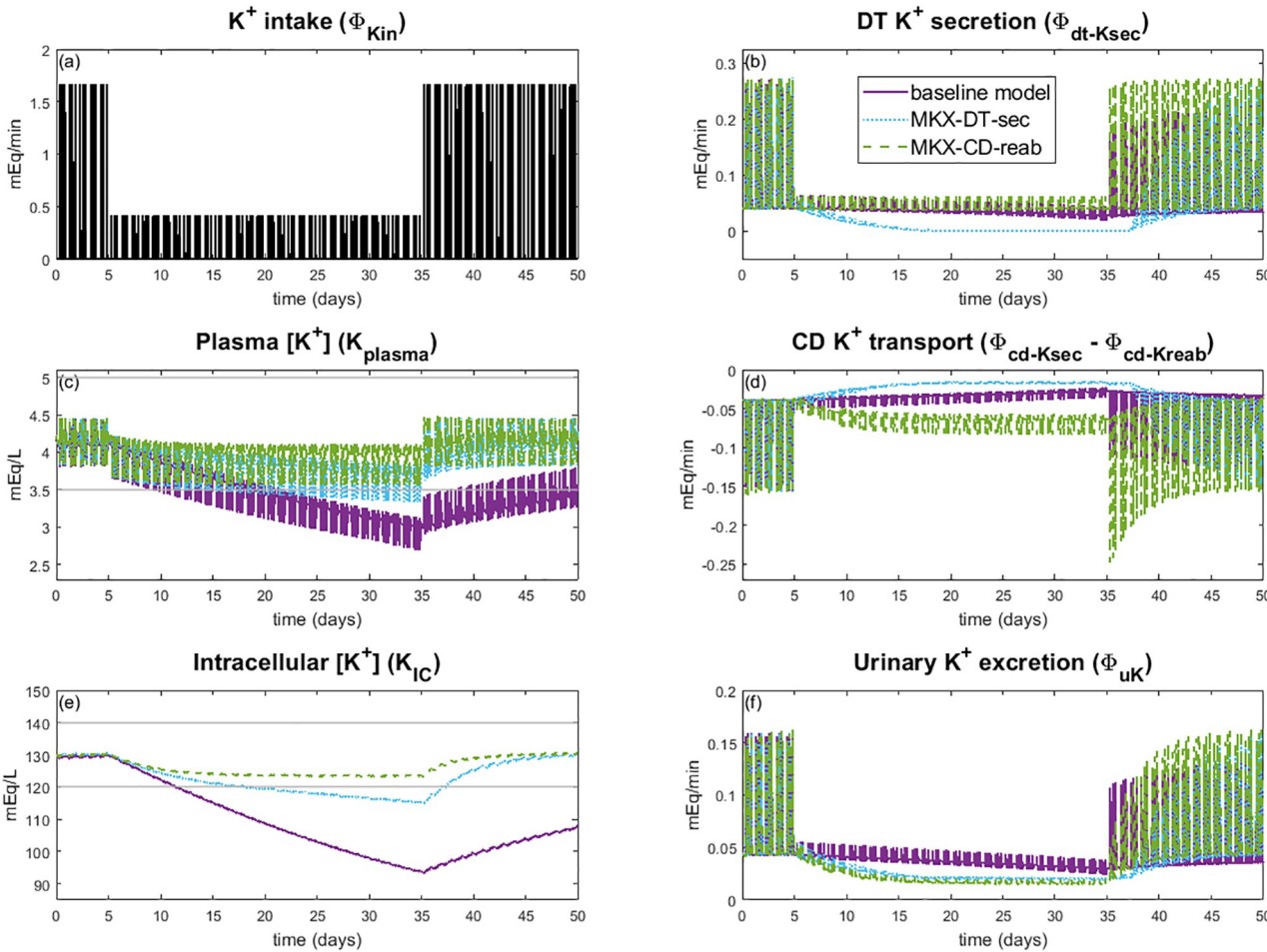

**Fig 9. Impact of muscle-kidney cross talk on K⁺ depletion.** Simulation results for K⁺ depletion experiments for the baseline model and muscle-kidney cross talk simulations (Case MKX-DT-sec and Case MKX-CD-reab). Potassium intake ($\Phi_{Kin}$) is the same for all three simulation types *(A)*. Note that Case MKX-CD-sec is not plotted since it had little impact from the baseline model results. Horizontal grey line shows normal range for plasma K⁺ concentration *(C)* and intracellular K⁺ concentration *(E)*. CD: collecting duct, DT: distal tubule

intracellular K⁺ concentration falls below normal range (Fig 9C and 9E). While plasma K⁺ concentration for Case MKX-DT-sec eventually falls below 3.5 mEq/L, the average plasma K⁺ concentration remains within normal range for the full K⁺ depletion experiment (Fig 9C). Intracellular plasma K⁺ concentration decreases substantially, but much slower than the base-line model simulations, reaching an average of 119 mEq/L at day 14 but only decreasing to 115 mEq/L at day 30 (Fig 9E). Both muscle-kidney cross talk simulations return to baseline plasma K⁺ and intracellular K⁺ concentrations within a few days of normal K⁺ intake, whereas the baseline model takes more than 15 days to recover normal levels. In summary, muscle-kidney cross talk makes the body more robust to periods of severe K⁺ depletion by substantially decreasing urine K⁺ excretion.

## Discussion

Maintaining extra- and intracellular K⁺ concentrations in the body within physiological ranges is crucial for proper cell function. The transmembrane K⁺ gradient is a major player in main-taining a proper membrane potential, which helps determine contractility of skeletal and

cardiac muscle [2]. Compared to other major electrolytes, the ratio of daily $K^+$ intake (about 70–120 mEq in a Western diet) to extracellular fluid amount (about 70 mEq) is the highest [1]. This high ratio means that maintaining the extracellular $K^+$ concentration within a tight normal range is a daily homeostatic challenge because of frequent periods of $K^+$ loading (e.g., a meal) and fasting. Mammals have evolved feedforward and feedback mechanisms to maintain a normal extracellular $K^+$ concentration by (i) ensuring a proper distribution of $K^+$ between the extra- and intracellular fluids and (ii) regulating urinary excretion of $K^+$. These mechanisms are complex and are not fully understood.

The model presented in this study includes a comprehensive description of the regulatory mechanisms involved in $K^+$ handling. In particular, represented here is the gastrointestinal feedforward control, in which higher dietary $K^+$ intake stimulates distal tubule $K^+$ secretion. Additionally, the model represents a feedback control, in which elevated plasma $K^+$ concentration stimulates ALD secretion to elevate renal tubular $K^+$ secretion and excretion, and vice versa. The model also includes the impact of both ALD and insulin secretion on $K^+$ sequestration into the skeletal muscle.

Model simulations were conducted to predict the impact of the individual feedback and feedforward mechanisms on $K^+$ balance. We predict that the gastrointestinal feedforward mechanism plays the largest role in regulating total body $K^+$ in the long term by stimulating distal tubule $K^+$ secretion during times of $K^+$ loading (Fig 7). Additionally, since typical meals contain glucose and thus stimulate insulin secretion, insulin's impact on $Na^+$-$K^+$-ATPase uptake likely evolved to store extra $K^+$ in the skeletal muscles to later be secreted back into the extracellular fluid (Fig 7C). While the model predicts that the insulin signal only has a small impact on plasma $K^+$ concentration, it is likely that during periods of $K^+$ loading, the slight increase in $Na^+$-$K^+$-ATPase uptake plays an important role in ensuring extra $K^+$ is stored in the intracellular fluid. ALD affects both the uptake of $K^+$ by $Na^+$-$K^+$-ATPase as well as targets $K^+$ secretion in the distal segments of the nephron. In this way, during low extracellular $K^+$ concentrations, secretion is decreased in the nephrons and uptake by $Na^+$-$K^+$-ATPase decreased so that $K^+$ remains in the extracellular fluid. Model simulations predict that the ALD feedback stabilizes plasma $K^+$ concentration (Fig 7B).

It has been shown that after a period of high $K^+$ loading, plasma $K^+$ concentrations return to normal almost immediately after the high $K^+$ intake is discontinued [38]. In contrast, the baseline model yields a markedly slower excretion of the excess $K^+$, with plasma and intracellular $K^+$ concentration remaining significantly above baseline after four days (Fig 8C and 8E). Recovery is accelerated with the addition of a hypothesized regulatory process: muscle-kidney cross talk, by which variations in intracellular $K^+$ concentration induce adjustments in renal $K^+$ transport. That cross talk enhances urine $K^+$ excretion to stabilize plasma and intracellular $K^+$ concentration in face of a massive $K^+$ load. In a period of $K^+$ depletion, without the cross talk plasma and intracellular $K^+$ concentration quickly decreased below normal ranges (Fig 9C and 9E) and remained low even after normal $K^+$ intake resumed. With the addition of a muscle-kidney cross talk signal, urine $K^+$ excretion was significantly suppressed (Fig 9C and 9E), resulting in much more stable plasma and intracellular $K^+$ concentration. These results suggest that muscle-kidney cross talk may significantly improve recovery times and prevent hyper- and hypokalemia. That said, it is prudent to note that conclusive data on a muscle-kidney cross talk signal has not yet emerged. Furthermore, even if muscle-kidney cross talk exists, it remains unclear which tubular segment(s) the signal acts on. Simulation results suggest that an effect on distal tubule $K^+$ secretion or collecting duct $K^+$ reabsorption is likely to impact urine $K^+$ excretion enough to sufficiently handle a major $K^+$ load or depletion.

## Model limitations and future work

Regulation of $K^+$ transport is coupled to other solutes. In particular, the secretion of $K^+$ along the distal tubules depends in part on $Na^+$ reabsorption. $Na^+$ regulation is not represented explicitly in this study, because our goal is to explore the impact of feedforward and feedback effects on $K^+$ homeostasis in isolation. The present model can be extended to incorporate $Na^+$ regulation. The resulting integrative model can be used to study the synergistic roles of $Na^+$ and $K^+$ in blood pressure regulation [39–41].

Another limitation is the uncertainty of the muscle-kidney cross talk hypothesis. Our baseline model does not include such mechanism, but without it the model was unable to capture normal $K^+$ regulation when significant $K^+$ loading or depletion was added. There are other mechanisms that impact $K^+$ homeostasis that have not been captured in the models presented in this study. One such mechanism is circadian rhythms which are known to impact secretion of hormones (such as ALD) as well as impact renal function [42–45]. To incorporate circadian rhythms, some model parameters, e.g., $\Phi_{GFR}$, would need to be made time dependent Progesterone has also shown to play a role in $K^+$ regulation in both males and females [46, 47]. Future work may study the impact of other regulatory mechanisms, as these mechanisms become better characterized by new experimental data.

Sex differences have been reported in renal $K^+$ handling [24–26], $Na^+$ balance [48–50], ALD concentration [27, 51], and plasma $K^+$ concentration [28], all of which affect $K^+$ homeostasis. The current model parameters were determined for a male human. Sex-specific $K^+$ models will need to be developed to capture female-specific $K^+$ regulation. That would require a better characterization of the effect of sex on model components and overall behaviors, and to incorporate those effects by setting sex-specific model parameter values. Additionally, pregnancy is known to alter $K^+$ regulation, leading to $K^+$ retention, to deliver sufficient $K^+$ to the rapidly developing fetus [52–55]. Female-specific mathematical models may be used to study the unique physiological state of pregnancy [56]. Another unique state of $K^+$ retention is in neonates and early childhood [57, 58]. These altered $K^+$ handling states may benefit from investigation through mathematical modeling.

A more comprehensive and inclusive model of $K^+$ homeostasis can be utilized to investigate $K^+$-related diseases and treatments. Hypokalemia and hyperkalemia are common electrolyte disorders caused by changes in $K^+$ intake, altered excretion, or transcellular shifts. Diuretic use and gastrointestinal losses are common causes of hypokalemia, whereas kidney disease, hyperglycemia, and medication use are common causes of hyperkalemia. Both hypo- and hyperkalemia can be potentially fatal [4, 5, 59]. Why does substantial $K^+$ imbalance emerge in some patients but not others? Simulations of a cohort of virtual patients by sampling selected model parameters for some distribution (e.g., Ref. [60]) may yield insights for predicting individual risks of dyskalemia.

## Supporting information

**S1 Text. Supplementary text file.**
(PDF)

**S1 Fig. Individual effects of feedforward and feedback controls.** Simulation results for plasma $K^+$ concentration *(B)* and intracellular $K^+$ concentration *(C)* for 10 days of normal $K^+$ intake *((A)*;100 mEq/day, 3 typical meals) for the baseline model, all control mechanisms off except the ALD effect on $Na^+$-$K^+$-ATPase uptake ($\rho_{al}$), all control mechanisms off except the gastrointestinal feedforward mechanism, all control mechanisms off except insulin ($\rho_{insulin}$),

and baseline model with all feedback effects off. Grey lines indicate normal range for $K_{plasma}$ and $K_{IC}$.
(TIF)

**S2 Fig. Impact of muscle-kidney cross talk on K⁺ loading.** The same simulation results as shown in Fig 8 of manuscript. Simulation results for K⁺ loading experiments for the baseline model and muscle-kidney cross talk simulations (Case MKX-DT-sec and Case MKX-CD-reab). Potassium intake *(A)* is the same for all three simulation types. Note that Case MKX-CD-sec is not plotted since it had little impact from the baseline model results. Horizontal grey line shows normal range for plasma K⁺ concentration *(C)* and intracellular K⁺ concentration *(E)*. CD: collecting duct, DT: distal tubule.
(TIF)

**S3 Fig. Impact of muscle-kidney cross talk parameter $m_{Kic}$ on K⁺ loading.** Simulation results for K⁺ loading experiments for the baseline model and distal tubule K⁺ secretion muscle-kidney cross talk simulations (Case MKX-DT-sec) for varied values of the parameter $m_{Kic}$. Potassium intake is the same for all simulation types. Horizontal grey line shows normal range for plasma K⁺ concentration *(A)* and intracellular K⁺ concentration *(B)*.
(TIF)

**S4 Fig. Impact of muscle-kidney cross talk parameter $m_{Kic}$ on K⁺ depletion.** Simulation results for K⁺ depletion experiments for the baseline model and distal tubule K⁺ secretion muscle-kidney cross talk simulations (Case MKX-DT-sec) for varied values of the parameter $m_{Kic}$. Potassium intake is the same for all simulation types. Horizontal grey line shows normal range for plasma K⁺ concentration *(A)* and intracellular K⁺ concentration *(B)*.
(TIF)

## Acknowledgments

We thank Karolina Suszek for illustrating the nephron in Fig 1.

## Author Contributions

**Conceptualization:** Anita T. Layton.

**Formal analysis:** Melissa M. Stadt, Anita T. Layton.

**Investigation:** Melissa M. Stadt, Jessica Leete, Sophia Devinyak.

**Methodology:** Melissa M. Stadt, Jessica Leete, Anita T. Layton.

**Software:** Melissa M. Stadt, Jessica Leete, Sophia Devinyak.

**Supervision:** Anita T. Layton.

**Writing – original draft:** Melissa M. Stadt, Jessica Leete.

**Writing – review & editing:** Melissa M. Stadt, Anita T. Layton.

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
