## [Decision Letter · Decision Letter 0]

26 Oct 2022

Dear Ms. Stadt,

Thank you very much for submitting your manuscript "A mathematical model of potassium homeostasis: Effect of feedforward and feedback controls" for consideration at PLOS Computational Biology.

As with all papers reviewed by the journal, your manuscript was reviewed by members of the editorial board and by several independent reviewers. In light of the reviews (below this email), we would like to invite the resubmission of a significantly-revised version that takes into account the reviewers' comments.

We cannot make any decision about publication until we have seen the revised manuscript and your response to the reviewers' comments. Your revised manuscript is also likely to be sent to reviewers for further evaluation.

Sincerely,

Daniel A Beard

Section Editor

PLOS Computational Biology

Daniel Beard

Section Editor

PLOS Computational Biology

Reviewer's Responses to Questions

**Comments to the Authors:**

Reviewer #1: In this study the authors describe a compartmental model of whole-body K+ regulation, and investigate the effect of various regulatory mechanisms on K+ homeostasis. The manuscript is well structured and clearly written. Moreover, I compliment the authors on the very interesting work, establishing a model of great value providing nice theoretical insights on a putative mechanism (e.g. muscle kidney crosstalk) that could contribute to K+ homeostasis under high K+ loading or depletion. However, my main concern is that some of the predictions may be dependent on the modeling assumptions and parameter values (which is always the case so not a problem per se), which are currently not shown. The sensitivity analysis is only done on the baseline model, not on the cases with(out) particular regulatory mechanisms and/or extreme K+ loading/depletion. As such, in addition to some more concrete question below, I think showing the parameter sensitivity to those simulation results as well would benefit the manuscript.

Questions/comments:

Are there K+ related pathologies/diseases? Did/could you explore those with the model?

Figure 1: if you add 2/3 + 25%+3%+10% > 100%?

I do not understand? Consider specifying a range per segment.

Out of curiosity – how does the gastrointestinal feedforward control mechanism work? (mechanistically) through which signaling networks is the gut linked to the kidney? Same for the putative muscle-kidney crosstalk. Could you elaborate/speculate?

How were the parameters determined?

Are these unique solutions? Could similar results be obtained with other (yet unidentified) feedback mechanisms? If so, how certain can we be of the current results?

Fig 2 – for completeness: add GFR and ALD explanation to the caption

“ As the filtrate passes through the nephrons, K+ (and other solutes and fluid not

represented in this model) is reabsorbed and returned to the general circulation along some tubular segments, and secreted into the lumen along the distal segments.” Since K+ is an ion & since some transporters are e.g. co-transporters; should the other solutes (or some of them) not also be considered; as they will also influence the K+ transport in the kidney?

What are γal × γKin in equation 14 – consider referring in the text to the aldosterone paragraph and equations 23-24. (similar for ρal × ρinsulin – link them better to the equations later on). Why multiply these effects and not add them?

“ We capture this effect by the scaling factor ρal (from Eq. 8), based on findings by Phakdeekitcharoen et al. [23].” Consider elaborating this more - on what is this fitted? What are the limitations/assumptions of these data (and thus the fit)?

“Rosic et al. [27] found that percent maximal Na+-K+-ATPase stimulation is a function of serum insulin concentration.” To help the reader the assumptions/limitations of this fit would be useful to add.

Where can we see the model fit? How well does the model fit the data?

Consider adding these figures to the supplement.

Table 2 – consider adding a column with measured experimental values, to see the how good the model matches reality.

It is not clear what the exact initial conditions are of Meal only”, “KCl only”, or “Meal + KCl”. Please clarify (e.g. with a table in the supplement).

Figure 4D – The model seems to underestimate the Kplasma for KCI only and overestimate for meal only – why is that? Which mechanisms might be missing in the model that could explaint his? Similarly for the urinary excretion (figure 5D) – the match for simulation results with the experimental data of the meal with KCl does not match well? Could you comment?

The feedforward loop (green) has an important influence in intracellular K+ (figure 7) – how dependent is this result on parameter values?

Figure 7 – it would be interesting to see how this figure is different in pathological states (so how the different feedback systems contribute (maybe worsen?) to K+ control.

Figure 7 – how do you explain that the order of the cases is not the same for plasma K+ and intracellular K+? E.g. for plasma the pink is highest, whereas for intracellular K+ the green is highest? Also – how come that the green line (only GI FF off) is higher than when all mechanisms are off (pink one)? Please clarify in the text.

Figure 8 – it would be good to have magnifications in the supplement (e.g. of a representative peak) – it is currently not easy to see the differences between the cases.

It would be good to elaborate the findings of the muscle-kidney crosstalk mechanisms, in particular, how do the parameters thereof influence the outcome, e.g. the omega?

Case MXK-DT-sec reduces distal tubule K+ secretion to almost 0 – is this physiological? By which transporters does this secretion occur? How would this affect the secretion of other solutes?

“Pettit and Vick [31] presented a two-compartment model to analyze data from dogs to understand the contribution of insulin to extrarenal K+ homeostasis. Their model represents K+ storage in the extra-

and intracellular compartments, together with their K+ exchange. A similar model was developed by Rabinowitz [32] to analyze data in sheep and understand the role of a gastrointestinal feedforward control mechanism. Youn et al. [33] used two-compartment and three-compartment models to analyze fluxes of K+ between the extracellular and intracellular space as well as K+ uptake by red blood cells. Maddah and Hallow [34] developed a quantitative systems pharmacology model that captures the effect of aldosterone on K+ homeostasis to simulate the effects of spironolactone treatment in patients with hyperaldosteronism.”

This paragraph would maybe fit better in the introduction as it describes the state of the art. Please elaborate what was learned from these models & which aspects were not modelled (the research gap) and which ones you addressed here?

If you prefer to leave it in the discussion, then I would extend the paragraph by comparing the predictions of these models with those presented in the manuscript and hypothesize on potential discrepancies.

The muscle kidney crosstalk is still hypothesized – what do we learn from the modeling predictions that can be used to do design some experiments to explore the muscle kidney crosstalk more/confirm it?

“ There are other mechanisms that impact K+ homeostasis that have not been captured in the models presented in this study. One such mechanism is circadian

rhythms which are known to impact secretion of hormones (such as ALD) as well as impact renal function [38–40]. Progesterone has also shown to play a role in K+ regulation in both males and females [41, 42].”

Could these mechanisms make that muscle-kidney crosstalk is not necessary (to help maintaining K+ balance)?

The discussion nicely lists some future perspectives – I would encourage the authors to specify what is needed to take those steps (is it just a matter of doing it or do we need more data, if so how to get that data etc)?

Reviewer #2: Well conceived project and well written manuscript.

A few minor corrections.

Typos and language:

Abstract, line 6: "homeostasis" instead of "regulation"?

Only outside Abstract, line 10: "K+ excretion" is typed twice..

Change last sentence of Summary: The significance of our "model is that it allows" to ......K+ "homeostasis" ..

Introduction, line 30: delete "in"

Internal K+ balance, line 84: missing "is" before "excreted".

For consistency use "mEq" throughout the MS, instead of switching between mmol and mEq.

Reviewer #3: The authors have developed a novel compartmental model of whole-body K+ regulation to explore specific feedforward and feedback effects on K+ homeostasis. The authors present in this manuscript the use of this model to explore the impact of various feedback mechanisms under several perturbations to gain preliminary insight into the significance of muscle-kidney cross talk. The assumptions and limitations of the developed model are clearly described and discussed in the manuscript and the authors suggest several areas ripe for future study with a suitable extensions to the presented model. As such, the presented model sets the baseline that can serve as the basis for a variety of future studies while still being able to elucidate preliminary insights to guide future studies.

The manuscript is very well written and clearly describes each of the simulation protocols used in the presented model analysis. It is great to see the authors putting a lot of effort into ensuring physical units are comprehensively presented throughout the model, although I did struggle with the mix of concentrations in mEq/L, ng/L, ng/dl, pmol/L, etc... - while I suspect that is just cause I am not used to seeing the Eq units, it would be nice to use consistent units throughout.

The authors should be commended for making their model implementation available on Github and I was very happy to be able to reproduce simulation results presented in the manuscript - thanks so much for providing a README that guides users through the various Matlab files. I would recommend that the authors tag the version of the code used to generate the specific results presented in the manuscript to ensure that future readers are able to unambiguously get to that version of the implementation. This could be achieved with a release on Github or publishing a version to zenodo or figshare or similar services. Also, given that this model could provide the basis for reuse in many future extensions, it would be good to be clear about the reusability of the code by adding a license under which the implementation is being made available on Github.

Minor revisions/suggestions

Line 360 - extra '.' after the Preston citation.

Line 469 - "100 mEq per da" should probably be "...per day"

Line 502 - misplaced '('

Lines 525-531 - MXK should probably be MKX and DTT should be DT ?

Figure 9 caption - consistency in the use of "cross talk" vs "crosstalk"

**Have the authors made all data and (if applicable) computational code underlying the findings in their manuscript fully available?**

Reviewer #1: Yes

Reviewer #2: Yes

Reviewer #3: Yes

PLOS authors have the option to publish the peer review history of their article (what does this mean?). If published, this will include your full peer review and any attached files.

Reviewer #1: No

Reviewer #2: No

Reviewer #3: **Yes: **David Nickerson
---

## [Editor Report · Decision Letter 1]

7 Nov 2022

Dear Ms. Stadt,

Thank you very much for submitting your manuscript "A mathematical model of potassium homeostasis: Effect of feedforward and feedback controls" for consideration at PLOS Computational Biology.

As with all papers reviewed by the journal, your manuscript was reviewed by members of the editorial board and by several independent reviewers. In light of the reviews (below this email), we would like to invite the resubmission of a significantly-revised version that takes into account the reviewers' comments.

We cannot make any decision about publication until we have seen the revised manuscript and your response to the reviewers' comments. Your revised manuscript is also likely to be sent to reviewers for further evaluation.

Sincerely,

Daniel A Beard

Section Editor

PLOS Computational Biology

Daniel Beard

Section Editor

PLOS Computational Biology
---

## [Decision Letter · Decision Letter 2]

28 Nov 2022

Dear Ms. Stadt,

We are pleased to inform you that your manuscript 'A mathematical model of potassium homeostasis: Effect of feedforward and feedback controls' has been provisionally accepted for publication in PLOS Computational Biology.

Please note that Reviewers 2 and 3 offer additional suggestions that are worth considering in making the final revisions of your paper.

Best regards,

Daniel A Beard

Section Editor

PLOS Computational Biology

Daniel Beard

Section Editor

PLOS Computational Biology

Reviewer's Responses to Questions

**Comments to the Authors:**

Reviewer #1: The authors rigorously addressed the comments (both in R1 and R2). I have no further comments, this is an interesting piece, valuable to the field.

Reviewer #2: This is a well researched and executed modelling study that provides useful new insights. The MS flows well, with exception of the second paragraph in the "Kidney function" section of the Introduction (lines 65-71) and a typo.

Lines 65-71: The confusion results from the introduction of "connecting tubule" and its function in the first sentence of this paragraph. The reasons are: Fig. 1 does not include "connecting tubule" and reference 8 distinguishes between early, aldsterone-insensitive connecting tubule (CNTe) and more distal aldosterone-sensitive connecting tubule (CNTas). Later on in the MS the "connecting tubule" is anyway subsumed under distal segment. Do the same in the Introduction.

My suggestion: 1) Eliminate the first sentence of the 2nd paragraph in "Kidney function" (line 65 +part of 66). 2) Add at the end of the current 3rd sentence ending in "distal tubular segments":, which include the connecting tubule.

Typo line 199: cotransporter or indirectly via membrane potential. (instead of directly via).

Reviewer #3: The revised manuscript addressed all the comments in original review - thanks! The only very minor point to note that the authors may want to consider is that the code published on Zenodo is under the CC-By license which is great for reusability, whereas the github repo itself has no license making it less reusable.

**Have the authors made all data and (if applicable) computational code underlying the findings in their manuscript fully available?**

Reviewer #1: Yes

Reviewer #2: Yes

Reviewer #3: Yes

PLOS authors have the option to publish the peer review history of their article (what does this mean?). If published, this will include your full peer review and any attached files.

Reviewer #1: No

Reviewer #2: **Yes: **Ulrich Hopfer, MD, PhD, Professor Emeritus

Reviewer #3: **Yes: **David Nickerson

---

## [Editor Report · Acceptance letter]

5 Dec 2022

PCOMPBIOL-D-22-01420R2 

A mathematical model of potassium homeostasis: Effect of feedforward and feedback controls

Dear Dr Stadt,

I am pleased to inform you that your manuscript has been formally accepted for publication in PLOS Computational Biology. Your manuscript is now with our production department and you will be notified of the publication date in due course.

With kind regards,

Zsofi Zombor
